# Explicit Eigenvalue Regularization Improves Sharpness-Aware Minimization

**Haocheng Luo[1], Tuan Truong[2], Tung Pham[2], Mehrtash Harandi[1], Dinh Phung[1], Trung Le[1]**

[1]Monash University, Australia
[2]VinAI Research, Vietnam

haocheng.luo@monash.edu, v.tuantm27@vinai.io, v.tungph4@vinai.io,
mehrtash.harandi@monash.edu, dinh.phung@monash.edu, trunglm@monash.edu

## Abstract

Sharpness-Aware Minimization (SAM) has attracted significant attention for its effectiveness in improving generalization across various tasks. However, its underlying principles remain poorly understood. In this work, we analyze SAM's training dynamics using the maximum eigenvalue of the Hessian as a measure of sharpness and propose a third-order stochastic differential equation (SDE), which reveals that the dynamics are driven by a complex mixture of second- and third-order terms. We show that alignment between the perturbation vector and the top eigenvector is crucial for SAM's effectiveness in regularizing sharpness, but find that this alignment is often inadequate in practice, which limits SAM's efficiency. Building on these insights, we introduce Eigen-SAM, an algorithm that explicitly aims to regularize the top Hessian eigenvalue by aligning the perturbation vector with the leading eigenvector. We validate the effectiveness of our theory and the practical advantages of our proposed approach through comprehensive experiments. Code is available at https://github.com/RitianLuo/EigenSAM.

## 1 Introduction

Understanding the generalization of deep learning algorithms is one of the core challenges in modern machine learning. Overparameterization makes the loss landscape of neural networks highly non-convex, often featuring numerous global optima, while simple gradient-based algorithms surprisingly tend to find solutions that generalize well. A body of empirical and theoretical work suggests that the "flatness" or "sharpness" of the minima is a promising explanation for generalization (Hochreiter and Schmidhuber, 1997; Keskar et al., 2016; Dinh et al., 2017; Jiang et al., 2019; Xie et al., 2020; Liu et al., 2023b), and the implicit bias of optimization algorithms drives them toward flatter solutions, thereby ensuring good generalization (Blanc et al., 2020; Wen et al., 2022; Arora et al., 2022; Damian et al., 2022; Ahn et al., 2023; Tahmasebi et al., 2024).

Inspired by research on flatness and generalization, recent work by Foret et al. (2021) proposed Sharpness-Aware Minimization (SAM), a dual optimization method that perturbs parameters before performing gradient descent to enhance generalization performance by minimizing sharpness. Although SAM has demonstrated empirical success across various fields (Foret et al., 2021; Kaddour et al., 2022), theoretical analysis of the principles underlying its success remains limited. The work by Compagnoni et al. (2023) explains SAM's generalization advantage as an implicit minimization of the gradient norm, while Wen et al. (2022) suggests that SAM regularizes the Hessian spectrum near the minima manifold. However, existing theories are either somewhat simplified or rely on overly

38th Conference on Neural Information Processing Systems (NeurIPS 2024).

idealized assumptions, leading to a noticeable gap between theory and practice. This gap limits their ability to fully explain the advantages of SAM (see Appendix A or contemporaneous work Song et al. (2024) for empirical evidence).

In this paper, we consider a widely used measure of sharpness: the largest eigenvalue of the Hessian matrix (Lyu et al., 2022; Arora et al., 2022; Wen et al., 2022; Damian et al., 2022). We extend the previous PAC-Bayes theory to demonstrate the importance of this measure for generalization. Our main contribution is an in-depth analysis of the training dynamics of SAM, expanding on the second-order Stochastic Differential Equation (SDE) proposed by Compagnoni et al. (2023) and revealing that the complex third-order terms play a crucial role in shaping SAM's implicit bias. Under ideal conditions, where the perturbation vector aligns well with the top eigenvector of the Hessian matrix, these terms effectively reduce the sharpness of the loss function. However, our experiments show that this alignment does not hold in real-world settings, limiting SAM's ability to effectively regularize sharpness. Based on our theoretical findings and experimental observations, we propose a new algorithm, Eigen-SAM, which intermittently estimates the top eigenvector of the Hessian matrix and incorporates its component orthogonal to the gradient into the perturbation, enabling explicit regularization of the top Hessian eigenvalue.

We summarize our contributions as follows:

- We prove a new theorem (Theorem 3.1) to establish the relationship between the top eigenvalue and generalization error, building on the general PAC-Bayes theorem (Alquier et al., 2016).

- We propose a third-order SDE (Theorem 4.1) to model the dynamics of SAM. This approach achieves a lower approximation error compared to the previous second-order SDE by Compagnoni et al. (2023) and additionally infers a close relationship between perturbation-eigenvector alignment and sharpness reduction (Corollary 4.1.1).

- We introduce a novel algorithm, Eigen-SAM, based on our theoretical insights and experimental observations (Section 5). This method aims to enhance alignment between the perturbation vector and the top eigenvector, resulting in a more effective reduction of sharpness.

- We validate our theory and the effectiveness of the proposed algorithm through comprehensive experiments (Section 6).

## 2 Related work

**Theoretical understanding of SAM.** SAM has garnered widespread attention for its significant improvements in generalization performance; however, the theoretical analysis provided in the original paper (Foret et al., 2021) is limited. The authors only presented a PAC-Bayes generalization bound, which is effective only for 0-1 loss. Subsequently, Andriushchenko and Flammarion (2022) attempted to further understand the success of SAM by restricting the network structure to diagonal linear networks and providing an implicit bias for SAM. They also established the first convergence result for SAM. Bartlett et al. (2023) conducted a detailed study of SAM's dynamics for quadratic loss, suggesting that it oscillates between the two sides of the minimum in the direction of greatest curvature and drifts toward flatter minima. However, the assumption of quadratic or locally quadratic loss settings is not realistic for practical deep learning models. More recently, Wen et al. (2022) extended the analysis of SAM's dynamics to general loss functions, assuming that all global minima form a connected manifold. Given sufficient training time and infinitesimally small $\eta$ and $\rho$, they rigorously proved that SAM's dynamics would track the trajectory of a Riemannian flow with respect to sharpness, achieving the same sharpness-reducing effect. On a different front, Compagnoni et al. (2023) applied the continuous-time approximation framework from Li et al. (2017) to analyze SAM dynamics, concluding that SAM implicitly minimizes the norm of the gradient scaled by $\rho$.

**Continuous-time approximations for discrete algorithms.** Substantial research demonstrates that the trajectory of stochastic discrete iterations with decaying step sizes will ultimately follow the solution of certain Ordinary Differential Equations (ODEs) (Harold et al., 1997; Borkar et al., 2009; Duchi and Ruan, 2018). Further developments in understanding deep learning algorithms were made by Li et al. (2017), who proposed a general and rigorous mathematical framework for continuous-time approximations, deriving SDEs for SGD and its various variants. They also provided experimental

evidence supporting the reasonableness of continuous-time approximations in real-world models (Li et al., 2021). In this paper, we follow this mathematical framework, reusing some of its notations and definitions.

# 3 Preliminaries

## 3.1 Notations

We start by introducing the notation used throughout our paper. Let $\mathcal{S}$ denote the training set, sampled from the true data distribution $\mathcal{D}$. For a given mini-batch $\gamma$, we define the mini-batch loss as $f_\gamma(x)$ with parameters $x \in \mathbb{R}^d$. The generalization loss is defined as $f_\mathcal{D}(x) = \mathbb{E}_{\gamma \sim \mathcal{D}}[f_\gamma(x)]$, while the empirical loss is defined as $f_\mathcal{S}(x) = \mathbb{E}_{\gamma \sim \mathcal{S}}[f_\gamma(x)]$. Since we primarily analyze and discuss the empirical loss in this paper, we drop the dependency on $\mathcal{S}$ for simplicity, denoting the empirical loss as $f(x)$ when no ambiguity arises.

We use $\|\cdot\|$ to denote the Euclidean norm. The $k$-th order derivative of the loss $f$ at $x$ is denoted by $\nabla^k f(x)$, which is a symmetric $k$-tensor in $(\mathbb{R}^d)^{\otimes k}$ when $x \in \mathbb{R}^d$. We denote by $\lambda_1(\nabla^2 f(x))$ the largest eigenvalue of the Hessian matrix $\nabla^2 f(x)$ and $v_1(\nabla^2 f(x))$ its corresponding unit eigenvector, with $\|v_1(\nabla^2 f(x))\| = 1$. Additionally, we use $\nabla^3 f(x)(u, v) \in \mathbb{R}^d$ to represent the application of the symmetric 3-tensor $\nabla^3 f(x)$ along directions $u$ and $v$.

## 3.2 Sharpness-Aware Minimization

SAM (Foret et al., 2021) seeks flat minima by minimizing the perturbed loss:

$$\min_x \max_{\|\epsilon\| \leq 1} f(x + \rho\epsilon),$$

where $\rho$ is a predefined hyperparameter controlling the radius of the perturbation. Solving the inner maximization problem leads to $\epsilon^{SAM}(x) = \frac{\nabla f(x)}{\|\nabla f(x)\|}$. Differentiating the perturbed loss with respect to $x$, we get:

$$\nabla f(x + \rho\epsilon^{SAM}(x)) = \frac{d(x + \rho\epsilon^{SAM}(x))}{dx} \nabla f(x)|_{x+\rho\epsilon^{SAM}(x)}$$
$$\approx \nabla f(x)|_{x+\rho\epsilon^{SAM}(x)}.$$

In the last approximation, Foret et al. (2021) ignore the dependency of $\epsilon^{SAM}(x)$ on $x$, leading to faster computational efficiency and higher generalization performance. Applying SAM in the stochastic case, the SAM iteration for mini-batch $\gamma_k$ is summarized as:

$$x_{k+1} = x_k - \eta\nabla f_{\gamma_k}\left(x_k + \rho\frac{\nabla f_{\gamma_k}}{\|\nabla f_{\gamma_k}\|}\right). \tag{1}$$

We use $\nabla f_{\gamma_k}(\cdot)$ and $\nabla f_{\gamma_k}$ to distinguish between those needing and not needing backpropagation of gradients, matching the algorithm in practice. Thus, the perturbation vector of mini-batch SAM can be written as:

$$\epsilon_\gamma^{SAM} = \frac{\nabla f_\gamma}{\|\nabla f_\gamma\|}. \tag{2}$$

## 3.3 SDE approximation for SGD and SAM

Li et al. (2017) developed the following SDE to approximate discrete SGD:

$$dX_t = -\nabla f(X_t)dt + \sqrt{\eta}(\Sigma^{1,1}(X_t))^{\frac{1}{2}}dW_t,$$

where $W_t$ is standard Brownian motion. Compagnoni et al. (2023) applied this framework to analyze the dynamics of SAM, deriving the following second-order SDE for SAM:

$$dX_t = \left(-\nabla f(X_t) - \rho\mathbb{E}\left[\frac{\nabla^2 f_\gamma(X_t)\nabla f_\gamma}{\|\nabla f_\gamma\|_2}\right]\right)dt + \sqrt{\eta}\left(\Sigma^{1,1}(X_t) + \rho(\Sigma^{1,2}(X_t) + \Sigma^{1,2}(X_t)^\top)\right)^{\frac{1}{2}}dW_t, \tag{3}$$

where $\Sigma^{a,b}$ denotes the covariance matrix of the $a$-th and $b$-th terms in the Taylor expansion of the perturbed loss (see Appendix B for the full expression). We refer to Eq.3 as the second-order SDE since it includes up to second-order partial derivatives in both the drift and diffusion coefficients.

### 3.4 Choice of sharpness measure

In this paper, we use the largest eigenvalue $\lambda_1(\nabla^2 f(x))$ as a measure of sharpness, similar to prior works (Lyu et al., 2022; Arora et al., 2022; Wen et al., 2022; Damian et al., 2022). Geometrically, the top eigenvalue of the Hessian matrix at a given point represents the maximal curvature of the loss function along any direction. Moreover, it is closely related to the concept of sharpness (defined as the maximum perturbed loss difference) used by Foret et al. (2021) at the minima. Note that at the minima, $\nabla f(x) = 0$; thus,

$$
\max_{\|\epsilon\| \leq 1} f(x + \rho\epsilon) - f(x) \approx \max_{\|\epsilon\| \leq 1} \rho\epsilon^\top \nabla f(x) + \frac{\rho^2}{2}\epsilon^\top \nabla^2 f(x)\epsilon
$$
$$
= \frac{\rho^2 \lambda_1(\nabla^2 f(x))}{2}.
$$

Another possible choice is the trace of the Hessian matrix. However, the trace of the Hessian scales with the dimensionality of the parameters, complicating cross-model comparisons and limiting this measure's applicability.

We further establish a PAC-Bayes theorem that bounds the generalization error through the top eigenvalue of the Hessian matrix. This theorem is based on the general PAC-Bayes theorem Alquier et al. (2016) and applies to bounded losses, not limited to 0-1 loss as in the work of Foret et al. (2021); Zhuang et al. (2022).

**Theorem 3.1.** (Generalization Bound) *Assume that the loss function is bounded by L, and the third-order partial derivative of the loss function is bounded by C. Additionally, we assume $f_{\mathcal{D}}(x) \leq \mathbb{E}_{\epsilon \sim \mathcal{N}(0,\sigma^2 \mathbb{I}_d)} f_{\mathcal{D}}(x + \epsilon)$, as in Foret et al. (2021). For any $\delta \in (0, 1)$ and $\sigma > 0$, with a probability over $1 - \delta$ over the choice of $\mathcal{S} \sim \mathcal{D}^n$, we have*

$$
f_{\mathcal{D}}(x) \leq f_{\mathcal{S}}(x) + \frac{d\sigma^2}{2}\lambda_1\left(\nabla^2 f_{\mathcal{S}}(x)\right) + \frac{Cd^3\sigma^3}{6}
$$
$$
+ \frac{L}{2\sqrt{n}}\sqrt{d\log\left(1 + \frac{\|x\|^2}{d\sigma^2}\right) + O(1) + 2\log\frac{1}{\delta} + 4\log(n + d)}.
$$

*where n is the number of samples.*

We defer the proof to Appendix C. Theorem 3.1 suggests that minimizing the top eigenvalue of the Hessian matrix is crucial for improving generalization ability.

## 4 Third-order SDE reveals implicit regularization in SAM

In this section, we delve into the discussion and derivation of the third-order SDE continuous-time approximation for SAM. In Section 4.1, we present heuristic derivations that provide intuitive insights into our approach. Following this, Section 4.2 offers a formal third-order SDE approximation for SAM, establishing the mathematical rigor of our framework. Finally, in Section 4.3, we propose a corollary linking perturbation-eigenvector alignment with eigenvalue regularization, furthering our understanding of the implicit regularization effects inherent in SAM.

### 4.1 Heuristic derivations for the third-order SDE

We begin by examining the drift coefficient in Compagnoni et al. (2023) (Eq. 3):

$$
-\nabla f(X_t) - \rho\mathbb{E}\left[\frac{\nabla^2 f_\gamma(X_t)\nabla f_\gamma}{\|\nabla f_\gamma\|}\right] = -\nabla f(X_t) - \rho\nabla\mathbb{E}\|\nabla f_\gamma(X_t)\|,
$$

where the second term indicates that SAM penalizes trajectories with large loss gradients. However, this formulation does not reveal an implicit regularization effect concerning sharpness, specifically regarding the top eigenvalue of the Hessian matrix. Thus, understanding the implicit bias on the Hessian matrix requires a third-order Taylor expansion. The missing cubic term in the Taylor expansion is:

$$
\frac{\rho^2}{2}\mathbb{E}\left[\frac{\nabla^3 f_\gamma(X_t)(\nabla f_\gamma, \nabla f_\gamma)}{\|\nabla f_\gamma\|^2}\right] = \frac{\rho^2}{2}\nabla\mathbb{E}\left[\frac{\nabla f_\gamma^\top \nabla^2 f_\gamma(X_t)\nabla f_\gamma}{\|\nabla f_\gamma\|^2}\right].
$$

The equality holds because $\nabla f_\gamma$ is treated as a perturbation vector independent of $X_t$, as SAM's implementation does not involve differentiating with respect to it (See Section 3.2 for a detailed description of this part). This third-order term suggests that SAM employs an additional gradient measurement to compute a specific third derivative: the gradient of the second derivative along the direction of the gradient.

**Hessian-gradient alignment during training.** Recent findings indicate that during training, the gradient implicitly aligns with the top eigenvector of the Hessian matrix under certain conditions: (1) training with normalized full-batch gradient descent (Arora et al., 2022); (2) a locally quadratic loss landscape (Bartlett et al., 2023); (3) training with SAM when very close to the minimizer manifold (Wen et al., 2022). This alignment phenomenon is crucial for interpreting the third-order term in our drift coefficient. Specifically, when the gradient is highly aligned with the top eigenvector, we have:

$$\frac{\rho^2}{2}\mathbb{E}\left[\frac{\nabla^3 f_\gamma(X_t)(\nabla f_\gamma, \nabla f_\gamma)}{\|\nabla f_\gamma\|^2}\right] \approx \frac{\rho^2}{2}\mathbb{E}\nabla^3 f_\gamma(X_t)(v_1(\nabla^2 f_\gamma(X_t)), v_1(\nabla^2 f_\gamma(X_t)))$$

$$= \frac{\rho^2}{2}\mathbb{E}\nabla\lambda_1(\nabla^2 f_\gamma(X_t)),$$

where the final equality follows from the properties of differentiating eigenvalues (see Magnus (1985) for a detailed discussion).

If this alignment phenomenon holds, we can conclude that the implicit bias of the drift coefficient aligns with the gradient of the top eigenvalue of the Hessian, thereby implicitly minimizing sharpness.

## 4.2   Formal third-order SDE approximation for SAM

In this subsection, we present the general formulation of the SDE for SAM. We refer to our SDE (Eq. 4) as the third-order SDE, to distinguish it from the second-order SDE (Eq. 3). For the complete statements and proofs, we refer the reader to Appendix B.

**Theorem 4.1.** (Third-order SDE for SAM, Informal Statement of Theorem B.4) *Let $0 < \eta < 1, T > 0$, $N = \lfloor T/\eta \rfloor$, and $\{x_k : k \geq 0\}$ denote the sequence of discrete SAM iterations defined by Eq. 1. Define $\{X_t : t \in [0,T]\}$ as the stochastic process satisfying the SDE*

$$dX_t = -\nabla\widetilde{f}^{SAM}(X_t)dt + \sqrt{\eta}(\Sigma^{SAM}(X_t))^{\frac{1}{2}}dW_t, \quad X_0 = x_0 \tag{4}$$

*with $\widetilde{f}^{SAM}(X_t) := f(X_t) + \rho\mathbb{E}\|\nabla f_\gamma(X_t)\| + \frac{\rho^2}{2}\mathbb{E}\frac{\nabla f_\gamma^\top \nabla^2 f_\gamma(X_t)\nabla f_\gamma}{\|\nabla f_\gamma\|^2}$,*

$$\Sigma^{SAM}(X_t) := \Sigma^{1,1}(X_t) + \rho(\Sigma^{1,2}(X_t) + \Sigma^{1,2}(X_t)^\top) + \rho^2\left(\Sigma^{2,2}(X_t) + \frac{1}{2}(\Sigma^{1,3}(X_t) + \Sigma^{1,3}(X_t)^\top)\right),$$

*where $\Sigma^{a,b}$ denotes the covariance matrix of the $a$-th and $b$-th terms in the Taylor expansion of the perturbed loss (see Appendix B for the full expression).*

*Under sufficient regularity conditions, let $\rho = \mathcal{O}(\eta^{\frac{1}{3}})$. Then, $\{X_t : t \in [0,T]\}$ is an order-1 weak approximation of $\{x_k : k \geq 0\}$, i.e., for any test function $g$ of at most polynomial growth, there exists a constant $C$ independent of $\eta$ such that*

$$\max_{k=0,1,\dots,N}|\mathbb{E}g(x_k) - \mathbb{E}g(X_{k\eta})| \leq C\eta.$$

Our proof relies on the third-order Taylor expansion of $f_{\gamma_k}\left(X_t + \rho\frac{\nabla f_{\gamma_k}}{\|\nabla f_{\gamma_k}\|}\right)$, carefully matching the first- and second-order conditional moments and quantifying the errors for higher-order terms. Our third-order SDE reveals that SAM's implicit bias includes a complex combination of second-order and third-order terms, with scales of $\rho$ and $\frac{\rho^2}{2}$, respectively. Compared to Compagnoni et al. (2023), our theorem offers two main advantages: first, we allow $\rho$ to take larger values ($\eta^{\frac{1}{3}}$ compared to $\eta^{\frac{1}{2}}$ in Compagnoni et al. (2023)), which is more consistent with real-world settings; equivalently, our SDE achieves a lower approximation error for a fixed $\rho$. Second, our SDE explicitly captures SAM's implicit bias on the Hessian matrix, manifesting as the gradient of the Hessian in the gradient's direction. Additionally, the diffusion coefficient in Eq. 4 implies that SAM injects additional noise in the form of the covariance of the higher-order terms in the Taylor expansion of the perturbed loss. This curvature-dependent noise aligns with recent studies (Gatmiry et al., 2024a,b), which demonstrate that label noise in SGD exhibits similar behavior to SAM.

### 4.3 Perturbation-eigenvector alignment and eigenvalue regularization

The implicit bias introduced by the third term in the drift coefficient of the SDE (Eq. 4), i.e., $\frac{\rho^2}{2}\mathbb{E}\frac{\nabla^3 f_\gamma(X_t)(\nabla f_\gamma, \nabla f_\gamma)}{\|\nabla f_\gamma\|^2}$, remains difficult to understand. As discussed in Section 4.1, if we assume that the perturbation vector is well-aligned with the top eigenvector, we can interpret the cubic term as the gradient of the top eigenvalue, leading to an implicit bias that decreases sharpness. Next, we quantify and formalize this heuristic approach. Define $\text{Align}(\epsilon, v_1) := 1 - \min_{s \in \{\pm 1\}} \|\frac{\epsilon}{\|\epsilon\|} - s \cdot v_1\|$ as a measure of alignment between the perturbation vector $\epsilon$ and the top eigenvector $v_1$. It is worth noting that $+v_1$ and $-v_1$ are equivalent eigenvectors, which is why we define alignment as the maximum over both $+v_1$ and $-v_1$.

**Corollary 4.1.1.** *Recall that $\epsilon_\gamma^{SAM}$ is defined in Eq. 2. Let $s^*$ denote the direction scalar, i.e., $s^* = \arg\min_{s \in \{\pm 1\}} \|\frac{\epsilon}{\|\epsilon\|} - s \cdot v_1\|$. Under the same conditions as in Theorem 4.1, and assuming a positive eigenvalue gap (see Assumption B.2 for a definition), we have the following:*
*1. If $\text{Align}(\epsilon_\gamma^{SAM}, v_1(\nabla^2 f_\gamma(X_t))) \geq 1 - \mathcal{O}(\rho)$, then the SDE (Eq. 4) becomes*

$$dX_t = -\nabla \widetilde{f}_\rho^{SAM}(X_t)dt + \sqrt{\eta}(\Sigma^{SAM})^{\frac{1}{2}}dW_t, \tag{5}$$

*where $\nabla \widetilde{f}_\rho^{SAM}(X_t) := \nabla f(X_t) + \rho\nabla\mathbb{E}\|\nabla f_\gamma(X_t)\| + \frac{\rho^2}{2}\nabla\mathbb{E}\lambda_1(\nabla^2 f_\gamma(X_t))$;*

*2. If $\text{Align}(\epsilon_\gamma^{SAM}, v_1(\nabla^2 f_\gamma(X_t))) \geq 1 - \mathcal{O}(\rho^2)$, then the SDE (Eq. 4) becomes*

$$dX_t = -\nabla \widetilde{f}_{\rho^2}^{SAM}(X_t)dt + \sqrt{\eta}(\Sigma^{SAM})^{\frac{1}{2}}dW_t, \tag{6}$$

*where $\nabla \widetilde{f}_{\rho^2}^{SAM}(X_t) := \nabla f(X_t) + \rho\mathbb{E}\big[s^* \cdot \lambda_1(\nabla^2 f_\gamma(X_t))v_1(\nabla^2 f_\gamma(X_t))\big] + \frac{\rho^2}{2}\nabla\mathbb{E}\lambda_1(\nabla^2 f_\gamma(X_t))$.*

The proof is deferred to Appendix B. In Corollary 4.1.1, we rigorously formalize our intuition from Section 4.1. If the alignment is at least $1 - \mathcal{O}(\rho)$, we conclude that the SAM trajectory comprises three components: the gradient of the loss, the gradient of the gradient norm, and the gradient of the top eigenvalue, with respective scales $1$, $\rho$, and $\frac{\rho^2}{2}$. Under this well-aligned condition, the SAM trajectory minimizes the loss while implicitly regularizing both the gradient norm and sharpness. This demonstrates SAM's complex implicit bias, which is not solely influenced by second- or third-order terms, as summarized in previous work (Wen et al., 2022; Compagnoni et al., 2023). For empirical evidence supporting this observation, we refer readers to Appendix A.

Furthermore, if the alignment is at least $1 - \mathcal{O}(\rho^2)$, then the gradient of the gradient norm oscillates in the direction of the top eigenvector. Notably, Bartlett et al. (2023) derived a similar discrete SAM dynamic under specific conditions (Theorem 20), where the parameter trajectory oscillates in the direction of the top eigenvector while regularizing the leading eigenvalue. However, their conditions are stricter than ours, assuming that the parameters are already close to the minimum and requiring a specific initialization. When these conditions are met, they require an alignment of the gradient with the top eigenvector of at least $1 - \mathcal{O}(\eta\rho) = 1 - \mathcal{O}(\rho^4)$. In comparison, our SDE framework is more general.

**Comparison with Wen et al. (2022).** Wen et al. (2022) derived an implicit bias similar to our cubic term in the third-order SDE (Eq. 5) for SAM through the analysis of the Riemannian flow near the minimizer manifold. However, our work fundamentally differs from theirs. First, their theory requires a much longer training time $\Theta(\eta^{-1}\rho^{-2})$ compared to our $\Theta(\eta^{-1})$. Thus, our SDE corresponds to the initial phase of their analysis regarding time scale, during which they do not conclude any implicit bias. In contrast, our SDE (Eq. 5), which indicates that the implicit bias comprises three components with different scales, provides richer insights in this phase. Second, they require $\eta\ln(1/\rho)$ to be sufficiently small, causing $\rho$ to be exponentially smaller than $\eta$, whereas our theory accommodates a more practical range, $\rho = \mathcal{O}(\eta^{\frac{1}{3}})$.

## 5 Eigen-SAM: an explicit regularization method for the top eigenvalue of the Hessian

### 5.1 Failure of perturbation-eigenvector alignment in practice

Within the theoretical framework of Section 4, a natural question arises: *Is the perturbation-eigenvector alignment sufficient in practice for SAM to effectively minimize sharpness?* Unfortunately,

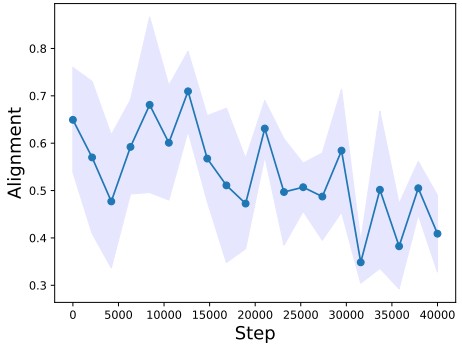 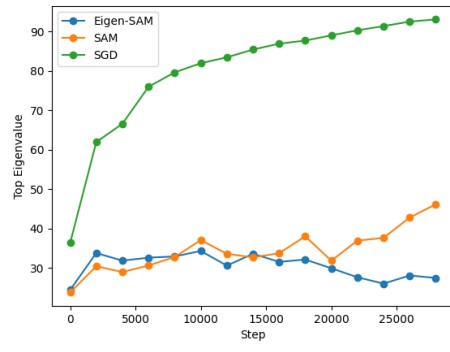

(a) Perturbation-eigenvector alignment    (b) Top eigenvalue of Hessian

Figure 1: Alignment and top eigenvalue for a 6-layer CNN model trained on CIFAR-10. The left panel shows the trend of alignment during SAM training; the shaded area represents the 95% confidence interval. The right panel displays the trend of the top eigenvalue over the course of training.

we have empirically verified that such alignment may be poor in practice, even for relatively simple models. As a result, the regularization effect on the largest eigenvalue, as discussed in Corollary 4.1.1, may not be clearly observable in practical scenarios.

To investigate this phenomenon, we trained a 6-layer SimpleCNN model, as used in Jastrzebski et al. (2021); Deng et al. (2024), on CIFAR-10 (Krizhevsky et al., 2009). Figure 1 illustrates two key findings from our experiments. The left panel shows that the alignment between the perturbation vector and the top eigenvector is indeed poor during training. Consequently, the right panel reveals that SAM is unable to efficiently minimize the top eigenvalue when alignment is weak. These results highlight the limitations of SAM in real-world scenarios where ideal alignment cannot be assumed.

## 5.2 Proposed method: Eigen-SAM

To address the issue of poor alignment, we propose a novel algorithm called Eigen-SAM, which aims to explicitly align the perturbation vector with the top eigenvector. This approach makes SAM's update closer to the SDE approximation in Eq. 5, where the third-order term in the drift coefficient can effectively minimize the largest eigenvalue. To achieve this alignment, we estimate the top eigenvector of the Hessian matrix once every $p$ mini-batch steps (with $p = 100$ in our implementation) using the power method for $q$ iterations (with $q = 5$ in our implementation). This strategy of intermittently estimating the Hessian matrix has been shown to be effective in practice (Liu et al., 2023a).

After obtaining an estimate $\hat{v}$ of the top eigenvector, we decompose it into components parallel and perpendicular to the gradient direction, then add the perpendicular component to the perturbation vector to enhance alignment explicitly:

$$\hat{v} = \hat{v}_{\parallel} + \hat{v}_{\perp} \tag{7}$$

$$\epsilon_{\gamma}^{\text{Eigen-SAM}} = \frac{\nabla f_{\gamma}}{\|\nabla f_{\gamma}\|} + \alpha \cdot \text{sign}(\langle \nabla f_{\gamma}, \hat{v} \rangle)\hat{v}_{\perp}, \tag{8}$$

where $\alpha$ is a hyperparameter that controls the strength of the explicit alignment. Since $+v$ and $-v$ are equivalent eigenvectors, we include $\text{sign}(\langle \nabla f_{\gamma}(x), \hat{v} \rangle)$ to determine the direction of $\hat{v}$. Here, we always choose $\hat{v}$ to have a smaller angle with the gradient. The full algorithm is presented in Algorithms 1 and 2. In Appendix D, we provide an in-depth discussion of the theoretical properties of Eigen-SAM, including sufficient conditions for improving alignment (Proposition D.1) and its convergence rate (Theorem D.2), which is comparable to that of SAM.

**Algorithm 1** Power iteration to estimate the top eigenvector

1: Initialize $\hat{v}$ as a random unit vector
2: **for** $t_2 = 1, 2, \ldots, q$ **do**
3:    Compute Hessian-vector product $\hat{v} = \nabla^2 f(x) \cdot \hat{v}$
4:    Normalize $\hat{v}$: $\hat{v} \leftarrow \hat{v}/\|\hat{v}\|$
5: **end for**
6: **return** $\hat{v}$

**Algorithm 2** Eigen-SAM

1: **for** $t_1 = 1, 2, \ldots, T$ **do**
2:    Compute mini-batch loss $f_\gamma(x_k)$
3:    **if** $t_1 \bmod p = 1$ **then**
4:       Use Algorithm 1 to estimate $\hat{v}$
5:    **end if**
6:    Compute the perturbation $\epsilon_t$ per Eq. 7 and 8
7:    Perturb the model: $\tilde{x}_t = x_k + \rho\epsilon_t$
8:    Update parameters: $x_{k+1} = x_k - \eta\nabla f_\gamma(\tilde{x}_k)$
9: **end for**
10: **return** $x_k$

## 5.3 Analysis of additional computational overhead

The additional overhead in Eigen-SAM arises from running the Hessian-vector product $q$ times every $p$ steps to estimate the top eigenvector. The Hessian-vector product requires roughly $1 - 2$ times the time needed to compute the gradient, so the overhead of our algorithm is approximately $2 + \frac{q}{p}$ to $2 + \frac{2q}{p}$ times that of SGD, compared to 2 times for standard SAM. For larger models, the computation time for the Hessian-vector product remains nearly constant. For a detailed analysis of the computation cost of Hessian-vector products, we refer readers to Dagréou et al. (2024).

# 6 Experiments

## 6.1 Numerical simulation of the third-order SDE

In this subsection, we validate the approximation error between our proposed SDE (Eq. 4) and the discrete SAM algorithm (Eq. 1). We trained a fully-connected network with one hidden layer, consisting of 784 hidden units and using GeLU activation, on the MNIST dataset (Deng, 2012). The training was conducted with $\eta = 0.01$ and $\rho = 0.2$ (where $\rho \approx \eta^{\frac{1}{3}}$). During training, we carefully tracked several key metrics, including training loss, test loss, test accuracy, parameter norm, gradient norm, and the top eigenvalue of the Hessian, as shown in Figure 2.

Our results demonstrate that the approximation error of our third-order SDE is significantly lower than that of the previous second-order SDE. Specifically, the curves of our SDE closely match those of the discrete SAM across all tracked metrics, underscoring the accuracy and reliability of our approximation. This close alignment suggests that our proposed continuous-time approximation provides a more precise representation of the discrete SAM dynamics, thus enhancing the theoretical understanding of SAM.

## 6.2 Image classification from scratch

To evaluate the effectiveness of Eigen-SAM, we applied it to several image classification tasks on benchmark datasets, including CIFAR-10 (Krizhevsky et al., 2009), CIFAR-100 (Krizhevsky et al., 2009), Fashion-MNIST (Xiao et al., 2017), and SVHN (Netzer et al., 2011). For these tasks, we used ResNet-18 (He et al., 2016), ResNet-50 (He et al., 2016), and WideResNet-28-10 (Zagoruyko and Komodakis, 2016) models.

We selected SGD as the base optimizer and applied basic data augmentation techniques, including horizontal flips, padding by four pixels, and random cropping. The batch size was set to 256, with training conducted for 200 epochs. We used an initial learning rate of 0.1 for CIFAR-10, Fashion-MNIST, and CIFAR-100, and 0.01 for SVHN, adjusting the learning rate over time with a cosine schedule. The weight decay was set to $5 \times 10^{-5}$, and the momentum was 0.9. Detailed hyperparameter settings are provided in Appendix E.

The test set performance, reported in Table 1 along with the $95\%$ confidence interval, shows that Eigen-SAM consistently achieves state-of-the-art performance across various datasets and models, validating its effectiveness and robustness.

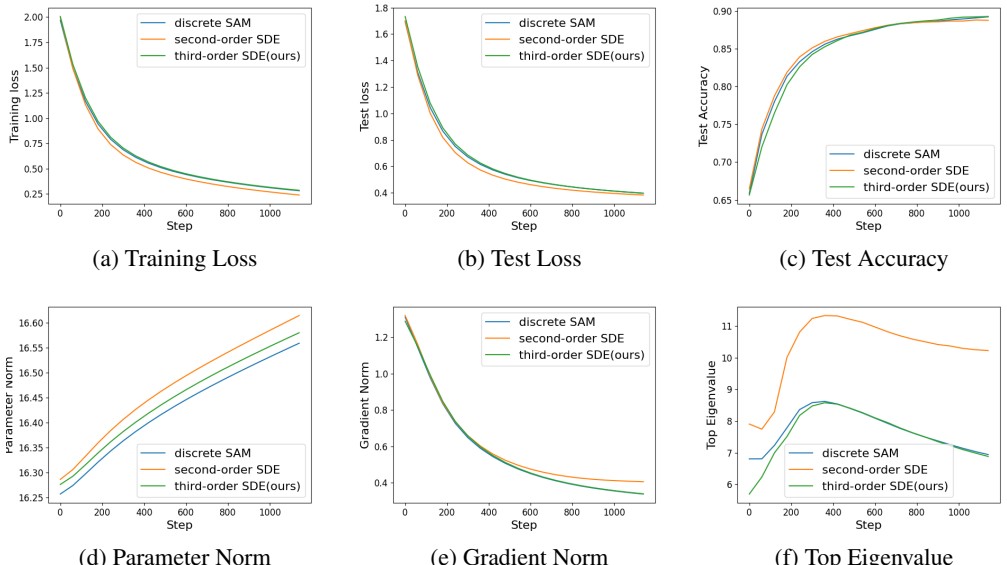

Figure 2: Training dynamics of discrete SAM, second-order SDE, and third-order SDE during training. Metrics include training loss, test loss, test accuracy, parameter norm, gradient norm, and the top Hessian eigenvalue. Each plot illustrates how each approach affects loss dynamics and key stability metrics.

Table 1: Test accuracy on CIFAR-10, CIFAR-100, Fashion-MNIST, SVHN.

| Architecture | Method | CIFAR-10 | CIFAR-100 | Fashion-MNIST | SVHN |
|---|---|---|---|---|---|
| ResNet18 | SGD | $94.8_{\pm 0.2}$ | $74.6_{\pm 0.2}$ | $94.9_{\pm 0.2}$ | $96.1_{\pm 0.1}$ |
| | SAM | $95.5_{\pm 0.1}$ | $77.4_{\pm 0.2}$ | $95.4_{\pm 0.1}$ | $96.3_{\pm 0.1}$ |
| | Eigen-SAM | $\mathbf{95.9}_{\pm 0.2}$ | $\mathbf{78.3}_{\pm 0.2}$ | $\mathbf{95.6}_{\pm 0.2}$ | $\mathbf{96.5}_{\pm 0.1}$ |
| ResNet50 | SGD | $95.0_{\pm 0.1}$ | $76.6_{\pm 0.2}$ | $94.8_{\pm 0.1}$ | $96.1_{\pm 0.1}$ |
| | SAM | $95.6_{\pm 0.2}$ | $79.0_{\pm 0.2}$ | $95.4_{\pm 0.1}$ | $96.4_{\pm 0.1}$ |
| | Eigen-SAM | $\mathbf{96.2}_{\pm 0.1}$ | $\mathbf{79.7}_{\pm 0.1}$ | $\mathbf{95.7}_{\pm 0.1}$ | $\mathbf{96.6}_{\pm 0.1}$ |
| WideResNet-28-10 | SGD | $95.7_{\pm 0.1}$ | $79.8_{\pm 0.2}$ | $95.1_{\pm 0.1}$ | $96.2_{\pm 0.1}$ |
| | SAM | $96.5_{\pm 0.1}$ | $82.0_{\pm 0.2}$ | $95.6_{\pm 0.1}$ | $96.4_{\pm 0.1}$ |
| | Eigen-SAM | $\mathbf{96.8}_{\pm 0.1}$ | $\mathbf{82.8}_{\pm 0.1}$ | $\mathbf{95.9}_{\pm 0.1}$ | $\mathbf{96.7}_{\pm 0.1}$ |

## 6.3 Finetuning

We evaluated performance by fine-tuning a ViT-B-16 model Dosovitskiy et al. (2020) pre-trained on ImageNet for CIFAR-10 and CIFAR-100. We used the checkpoint provided by PyTorch's official repository[1]. For SAM and Eigen-SAM, we used an initial learning rate of 0.01 and trained for 4k steps, while for SGD, we trained for 8k steps. Table 2 shows the test accuracy, where Eigen-SAM consistently outperforms the baselines.

Table 2: Test accuracy for fine-tuning ViT-B-16 pretrained on ImageNet-1K on CIFAR-10 and CIFAR-100.

| Architecture | Method | CIFAR-10 | CIFAR-100 |
|---|---|---|---|
| ViT-B-16 | SGD | $98.0_{\pm 0.1}$ | $88.6_{\pm 0.1}$ |
| | SAM | $98.4_{\pm <0.1}$ | $89.5_{\pm 0.1}$ |
| | Eigen-SAM | $\mathbf{98.5}_{\pm 0.1}$ | $\mathbf{89.8}_{\pm 0.1}$ |

---

[1]https://pytorch.org/vision/main/models/generated/torchvision.models.vit_b_16.html

### 6.4 Sensitivity analysis

We investigated the impact of varying the hyperparameter $\alpha$ on the test accuracy of Eigen-SAM. We conducted experiments on ResNet-18 with CIFAR-100, testing a range of $\alpha$ values, as shown in Figure 3. We observed that the test accuracy peaks at $\alpha = 0.2$, yielding a 0.9% improvement in test accuracy compared to $\alpha = 0$, which corresponds to the standard SAM. These results suggest that $\alpha$ is a robust hyperparameter, as its variations do not cause significant performance fluctuations, while consistently enhancing performance.

In Table 3 and Table 4 in Appendix F, we demonstrate how larger values of $p$ affect generalization performance and observe that setting $p$ to 1000 (resulting in less than 1% additional overhead) retains most of the performance gains. In Figure 5 in Appendix F, we demonstrate the convergence speed of Algorithm 1, which typically requires only a few steps to converge.

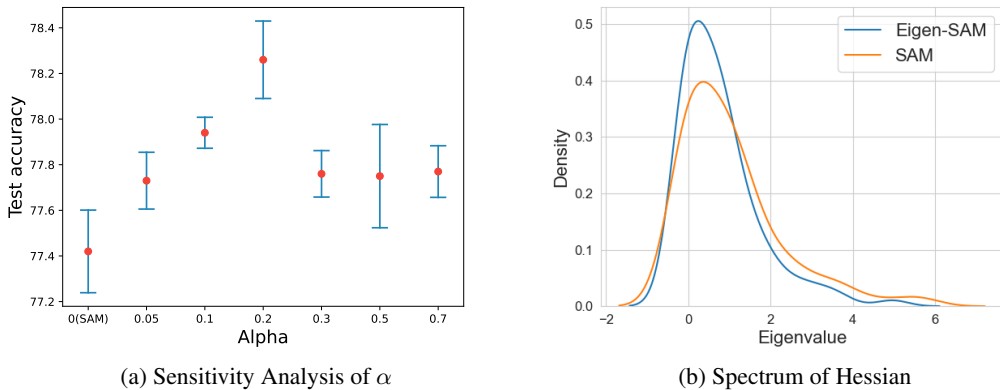

(a) Sensitivity Analysis of $\alpha$        (b) Spectrum of Hessian

Figure 3: Left: Sensitivity analysis of $\alpha$; the blue lines indicate the confidence interval. Right: Spectrum of the Hessian at the end of training.

### 6.5 Hessian spectrum

Figure 3 shows the Hessian spectrum at the 200th epoch for ResNet-18 trained on CIFAR-100 using SAM and Eigen-SAM. We observe that the model trained with Eigen-SAM has both a smaller top eigenvalue and trace, with more eigenvalues concentrated near zero. This observation aligns with our motivation for proposing Eigen-SAM and explains why Eigen-SAM generalizes better than SAM.

## 7 Conclusion

In this work, we analyzed the training dynamics of SAM using a third-order SDE, identifying the alignment between the perturbation vector and the top eigenvector as a crucial factor for effective sharpness regularization. However, our empirical analysis showed that this alignment is often poor in practice. Building on our theoretical framework and experimental insights, we proposed Eigen-SAM, an algorithm that intermittently estimates the top eigenvector of the Hessian matrix and incorporates its component orthogonal to the gradient into the perturbation, explicitly regularizing the top Hessian eigenvalue. Extensive experiments demonstrated that our third-order SDE yields a smaller approximation error than previous models and that Eigen-SAM achieves state-of-the-art performance across various tasks, validating both its accuracy and effectiveness.

## Acknowledgements

We would like to thank Zehang Deng for helpful discussions. We also appreciate the constructive feedback provided by the anonymous reviewers. This work was supported by ARC DP23 grant DP230101176 and by the Air Force Office of Scientific Research under award number FA2386-23-1-4044.

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

**Appendix**

## A  Additional empirical evidence

In this section, we present empirical evidence highlighting discrepancies between existing theories and practical observations. We demonstrate this by designing several "counterexample" algorithms and comparing their performance with SAM.

First, the theory developed by Wen et al. (2022) suggests that SAM's implicit regularization primarily occurs near the minima manifold, where $\nabla f(x) \approx 0$ and the quadratic term dominates the perturbed term. To test this, we designed a counterexample algorithm called Reverse-SAM, which applies SAM's ascent step using a negative normalized gradient, i.e., $\epsilon = -\frac{\nabla f_\gamma(x)}{\|\nabla f_\gamma(x)\|}$. If this theory holds in practice, then Reverse-SAM should perform similarly to SAM, as the sign of the perturbation vector would not affect the implicit bias induced by the quadratic term.

The second counterexample algorithm we consider is Explicit Gradient Regularization (EGR), a regularization method with a long history, tracing back to the works of Barrett and Dherin (2020) and Drucker and Le Cun (1991). The second-order SDE proposed by Compagnoni et al. (2023) suggests that SAM's implicit bias is equivalent to optimizing the objective $f(x) + \rho\|\nabla f(x)\|$. Therefore, if this second-order SDE model is accurate in practice, EGR with a regularization coefficient equal to $\rho$ should exhibit performance comparable to SAM.

We trained ResNet-18 on CIFAR-10 to compare the performance of these two counterexample algorithms with SAM. Figure 4 shows the training and test losses. Additionally, the test accuracies for SAM, EGR, and Reverse-SAM were $95.5\% \pm 0.1\%$, $95.0\% \pm 0.2\%$, and $94.4\% \pm 0.2\%$, respectively. Both counterexample algorithms show noticeable performance gaps from SAM across all metrics, highlighting the limitations of existing theories in explaining SAM's practical outcomes.

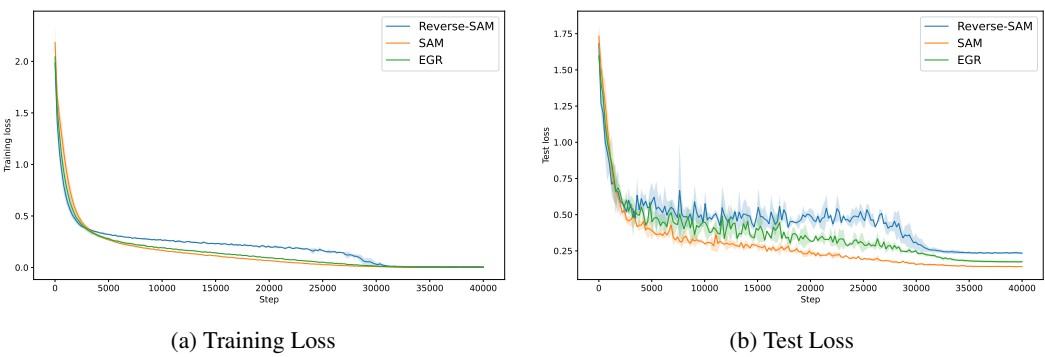

(a) Training Loss            (b) Test Loss

Figure 4: Comparison of training loss and test loss metrics across algorithms.

## B  Proofs for SDEs (Section 4)

We follow the notation of multi-index, which is commonly used in SDE literature:

- A multi-index is an n-tuple of non-negative integers $\alpha = (\alpha_1, \alpha_2 ..., \alpha_n)$
- $|\alpha| := \alpha_1 + \alpha_2 + ... + \alpha_n$
- $\alpha! := \alpha_1!\alpha_2!...\alpha_n!$
- For $x = (x_1, x_2 ..., x_n) \in \mathbb{R}^n$, $x^\alpha := x_1^{\alpha_1} x_2^{\alpha_2} ... x_n^{\alpha_n}$
- For a multi-index $\beta$, $\partial_\beta^{|\beta|} f(x) := \frac{\partial^{|\beta|}}{\partial x_1^{\beta_1} \partial x_2^{\beta_2} ... \partial x_n^{\beta_n}} f(x)$

We denote the partial derivative with respect to $x_i$ by $\partial_{e_i}$.

Following the SDE framework by Mil'shtein (1986); Li et al. (2017); Compagnoni et al. (2023), we have the following definitions and assumptions:

**Definition B.1.** *Let $G$ denote the set of continuous functions $\mathbb{R}^d \to \mathbb{R}$ of at most polynomial growth, i.e. $g \in G$ if there exists positive integers $\kappa_1, \kappa_2 > 0$ such that*

$$|g(x)| \le \kappa_1(1 + \|x\|^{2\kappa_2})$$

*for all $x \in \mathbb{R}^d$. Moreover, for each integer $\alpha \ge 1$ we denote by $G^\alpha$ the set of $\alpha$-times continuously differentiable functions $\mathbb{R}^d \to \mathbb{R}$ which, together with its partial derivatives up to and including order $\alpha$, belong to $G$.*

This definition originates from the field of numerical analysis of SDEs (Mil'shtein, 1986). In the case of $g(x) = \|x\|^j$, the bound restricts the difference between the $j$-th moments of the discrete process and those of the continuous process. We write $\mathcal{O}(\rho^{\alpha_1}\eta^{\alpha_2})$ to denote that there exists a function $K \in G$ independent with $\rho$, $\eta$, such that the error terms are bounded by $K\rho^{\alpha_1}\eta^{\alpha_2}$.

**Assumption B.1.** *Assume that the following conditions on $f$, $f_i$ and their gradients are satisfied:*

- $\nabla f$, $\nabla f_i$ *satisfy a Lipschitz condition: there exists $L > 0$ such that*

$$\|\nabla f(x) - \nabla f(y)\| + \sum_{i=1}^{n} \|\nabla f_i(x) - \nabla f_i(y)\| \le L\|x - y\|;$$

- $f$, $f_i$ *and its partial derivatives up to order 7 belong to $G$;*

- $\nabla f$, $\nabla f_i$ *satisfy a growth condition: there exists $M > 0$ such that*

$$\|\nabla f(x)\| + \sum_{i=1}^{n} \|\nabla f_i(x)\| \le M(1 + \|x\|);$$

- $g$ *and its partial derivatives up to order 6 belong to $G$;*

See Li et al. (2017) for a detailed discussion on Assumption B.1.

To prove the Corollary 4.1.1, we need an additional assumption to ensure that the top eigenvalue is differentiable. Note that this assumption is common in recent works (Damian et al., 2022; Wen et al., 2022) and easy to satisfy.

**Assumption B.2.** (Eigenvalues gap) *For all $k > 0$, $\lambda_1(\nabla^2 f(x_k)) > \lambda_2(\nabla^2 f(x_k))$, and $\lambda_1(\nabla^2 f(X_{k\eta})) > \lambda_2(\nabla^2 f(X_{k\eta}))$.*

In the proof of our main theorem, we will utilize the following two auxiliary lemmas.

**Lemma B.1.** (Lemma 1 (Li et al., 2017)) *Let $0 < \eta < 1$. Consider a stochastic process $\{X_t : t > 0\}$ satisfying the SDE*

$$dX_t = b(X_t)dt + \eta^{\frac{1}{2}}\sigma(X_t)dW_t$$

*with $X_0 = x \in \mathbb{R}^d$ and $b, \sigma$ together with their derivatives belong to $G$. Define the one-step difference $\Delta = X_\eta - x$, then we have*

$$\mathbb{E}\Delta_i = b_i\eta + \frac{1}{2}\left[\sum_{j=1}^{d} b_j\partial_{e_j}b_i\right]\eta^2 + \mathcal{O}\left(\eta^3\right) \quad \forall i = 1, \dots, d; \tag{9}$$

$$\mathbb{E}\Delta_i\Delta_j = \left[b_ib_j + \sigma\sigma_{(ij)}^T\right]\eta^2 + \mathcal{O}\left(\eta^3\right) \quad \forall i, j = 1, \dots, d; \tag{10}$$

$$\mathbb{E}\prod_{j=1}^{s}\Delta_{(i_j)} = \mathcal{O}\left(\eta^3\right) \quad \forall s \ge 3, i_j = 1, \dots, d. \tag{11}$$

*all the functions are evaluated at $x$.*

**Lemma B.2.** (Theorem 2 and Lemma 5 (Mil'shtein, 1986) )*Under Assumption B.1, we define the one-step difference for the stochastic process $\Delta = X_\eta - x$. if in addition there exists $K_1, K_2, K_3, K_4 \in G$*

*so that*

$$\left|\mathbb{E}\Delta_i - \mathbb{E}\bar{\Delta}_i\right| \leq K_1(x)\eta^2, \quad \forall i = 1, \ldots, d; \tag{12}$$

$$\left|\mathbb{E}\Delta_i\Delta_j - \mathbb{E}\bar{\Delta}_i\bar{\Delta}_j\right| \leq K_2(x)\eta^2, \quad \forall i,j = 1, \ldots, d; \tag{13}$$

$$\left|\mathbb{E}\prod_{j=1}^{s}\Delta_{i_j} - \mathbb{E}\prod_{j=1}^{s}\bar{\Delta}_{i_j}\right| \leq K_3(x)\eta^2, \quad \forall s \geq 3, \quad \forall i_j \in \{1, \ldots, d\}; \tag{14}$$

$$\mathbb{E}\prod_{j=1}^{3}\left|\bar{\Delta}_{i_j}\right| \leq K_4(x)\eta^2, \quad \forall i_j \in \{1, \ldots, d\}. \tag{15}$$

*Then, for each $g \in G$, there exists a constant $C$ such that*

$$\max_{k=0,1,\ldots N}\left|\mathbb{E}g(x_k) - \mathbb{E}g(X_{k\eta})\right| \leq C\eta$$

Next, we will prove a lemma that controls the moments of the discrete process for SAM:

**Lemma B.3.** *Under Assumption B.1, let $0 < \eta < 1$. We define:*

$$\partial_{e_i}\tilde{f}^{SAM}(x) := \partial_{e_i}f(x) + \rho\mathbb{E}\left[\frac{\sum_j \partial^2_{e_i+e_j}f_\gamma(x)\partial_{e_j}f_\gamma(x)}{\|\nabla f_\gamma(x)\|}\right] + \frac{\rho^2}{2}\mathbb{E}\left[\frac{\sum_{jk}\partial^3_{e_i+e_j+e_k}f_\gamma(x)\partial_{e_j}f_\gamma(x)\partial_{e_k}f_\gamma(x)}{\|\nabla f_\gamma(x)\|^2}\right].$$

*In addition, we define the one-step difference for the discrete-time algorithm as $\bar{\Delta} = x_1 - x$. Then we have:*

1. $\mathbb{E}\bar{\Delta}_i = -\partial_{e_i}\tilde{f}^{SAM}(x)\eta + \mathcal{O}\left(\eta\rho^3\right), \quad \forall i = 1, \ldots, d;$          (16)

2. $\mathbb{E}\bar{\Delta}_i\bar{\Delta}_j = \partial_{e_i}\tilde{f}^{SAM}(x)\partial_{e_j}\tilde{f}^{SAM}(x)\eta^2 + \Sigma^{SAM}_{(ij)}\eta^2 + \mathcal{O}\left(\eta^2\rho^3\right), \quad \forall i,j = 1, \ldots, d;$    (17)

3. $\mathbb{E}\prod_{j=1}^{s}\bar{\Delta}_{i_j} = \mathcal{O}\left(\eta^3\right), \quad \forall s \geq 3, \quad i_j \in \{1, \ldots, d\}.$                (18)

*All functions are evaluated at $x$.*

*Proof.* To evaluate $\mathbb{E}\bar{\Delta}_i = -\mathbb{E}\left[\partial_{e_i}f_\gamma\left(x + \frac{\rho}{\|\nabla f_\gamma(x)\|}\nabla f_\gamma(x)\right)\right]$, we start by analyzing $\partial_{e_i}f_\gamma\left(x + \frac{\rho}{\|\nabla f_\gamma(x)\|}\nabla f_\gamma(x)\right)$, which is the partial derivative in the direction $e_i$. Taking the Taylor expansion, we have:

$$\begin{aligned}
\partial_{e_i}f_\gamma\left(x + \frac{\rho}{\|\nabla f_\gamma(x)\|}\nabla f_\gamma(x)\right) =\ &\partial_{e_i}f_\gamma(x) + \sum_{|\alpha|=1}\partial^2_{e_i+\alpha}f_\gamma(x)\rho\frac{\partial_\alpha f_\gamma(x)}{\|\nabla f_\gamma(x)\|} \\
&+ \frac{1}{2}\sum_{|\alpha|=2}\partial^3_{e_i+\alpha}f_\gamma(x)\rho^2\left(\frac{\partial_\alpha f_\gamma(x)}{\|\nabla f_\gamma(x)\|}\right)^\alpha \\
&+ \mathcal{R}^{\partial_{e_i}f_\gamma(x)}_{x,1}\left(\rho\frac{\nabla f_\gamma(x)}{\|\nabla f_\gamma(x)\|}\right),
\end{aligned} \tag{19}$$

where the residual term is defined in Folland (2005). For some constant $c \in (0, 1)$, it holds that

$$\mathcal{R}^{\partial_{e_i}f_\gamma(x)}_{x,1}\left(\rho\frac{\nabla f_\gamma(x)}{\|\nabla f_\gamma(x)\|}\right) = \sum_{|\alpha|=3}\frac{\partial^4_{e_i+\alpha}f_\gamma\left(x + c\rho\frac{\nabla f_\gamma(x)}{\|\nabla f_\gamma(x)\|}\right)\rho^{|\alpha|}\left(\frac{\nabla f_\gamma(x)}{\|\nabla f_\gamma(x)\|}\right)^\alpha}{\alpha!}. \tag{20}$$

Now, we observe that

$$K_i(x) := \frac{\partial^4_{e_i+\alpha}f_\gamma\left(x + c\rho\frac{\nabla f_\gamma(x)}{\|\nabla f_\gamma(x)\|}\right)\rho^{|\alpha|}\left(\frac{\nabla f_\gamma(x)}{\|\nabla f_\gamma(x)\|}\right)^\alpha}{\alpha!}$$

is a finite sum of products of functions that, by assumption, are in $G$. We can rewrite Equation (19) as

$$\partial_{e_i} f_\gamma \left( x + \frac{\rho}{\|\nabla f_\gamma(x)\|} \nabla f_\gamma(x) \right) = \partial_{e_i} f_\gamma(x) + \sum_{|\alpha|=1} \partial^2_{e_i+\alpha} f_\gamma(x) \rho \frac{\partial_\alpha f_\gamma(x)}{\|\nabla f_\gamma(x)\|}$$
$$+ \frac{1}{2} \sum_{|\alpha|=2} \partial^3_{e_i+\alpha} f_\gamma(x) \rho^2 \left( \frac{\partial_\alpha f_\gamma(x)}{\|\nabla f_\gamma(x)\|} \right)^\alpha$$
$$+ \rho^3 K_i(x), \tag{21}$$

which implies that

$$\mathbb{E} \partial_{e_i} f_\gamma \left( x + \frac{\rho}{\|\nabla f_\gamma(x)\|} \nabla f_\gamma(x) \right) = \partial_{e_i} f(x) + \rho \mathbb{E} \left[ \frac{\sum_j \partial^2_{e_i+e_j} f_\gamma(x) \partial_{e_j} f_\gamma(x)}{\|\nabla f_\gamma(x)\|} \right]$$
$$+ \frac{\rho^2}{2} \mathbb{E} \left[ \frac{\sum_{jk} \partial^3_{e_i+e_j+e_k} f_\gamma(x) \partial_{e_j} f_\gamma(x) \partial_{e_k} f_\gamma(x)}{\|\nabla f_\gamma(x)\|^2} \right]$$
$$+ \rho^3 \bar{K}_i(x), \tag{22}$$

where $\bar{K}_i(x) = \mathbb{E} K_i(x)$.

Therefore, we have $\forall i = 1, 2, \ldots, d$,

$$\mathbb{E} \bar{\Delta}_i = -\partial_{e_i} \tilde{f}^{SAM}(x) \eta + \mathcal{O}\left( \eta \rho^3 \right).$$

To prove the second statement, by definition, we have

$$\text{Cov}(\bar{\Delta}_i, \bar{\Delta}_j) = \eta^2 \big( \Sigma^{1,1}_{i,j}(x) + \rho(\Sigma^{1,2}_{i,j}(x) + \Sigma^{1,2}_{i,j}(x)^\top)$$
$$+ \rho^2 (\Sigma^{2,2}_{i,j} + \frac{1}{2}(\Sigma^{1,3}_{i,j}(x) + \Sigma^{1,3}_{i,j}(x)^\top))) + \mathcal{O}(\eta^2 \rho^3) \tag{23}$$
$$= \eta^2 \Sigma^{SAM}(x) + \mathcal{O}(\eta^2 \rho^3). \tag{24}$$

$$\mathbb{E} \bar{\Delta}_i \bar{\Delta}_j = \mathbb{E} \bar{\Delta}_i \mathbb{E} \bar{\Delta}_j + \text{Cov}(\bar{\Delta}_i, \bar{\Delta}_j) \tag{25}$$
$$= \partial_{e_i} \tilde{f}^{SAM}(x) \partial_{e_j} \tilde{f}^{SAM}(x) \eta^2 + \eta^2 \Sigma^{SAM}(x) + \mathcal{O}(\eta^2 \rho^3). \tag{26}$$

Finally, it is clear that

$$\mathbb{E} \prod_{j=1}^s \bar{\Delta}_{i_j} = \mathcal{O}\left( \eta^s \right), \quad \forall s \geq 3, \quad i_j \in \{1, \ldots, d\} \tag{27}$$
$$= \mathcal{O}\left( \eta^3 \right), \quad \forall s \geq 3, \quad i_j \in \{1, \ldots, d\}. \tag{28}$$

$\square$

Now we are ready to state the theorem and prove it.

**Theorem B.4.** (Stochastic modified equations) *Under Assumption B.1, let $0 < \eta < 1, T > 0, N = \lfloor T/\eta \rfloor$. Let $x_k \in \mathbb{R}^d, 0 \leq k \leq N$ denote the sequence of SAM iterations defined by Equation 1. Define $\{X_t\}$ as the stochastic process satisfying the SDE*

$$dX_t = -\nabla \tilde{f}^{SAM}(X_t)dt + \sqrt{\eta}(\Sigma^{SAM}(X_t))^{\frac{1}{2}} dW_t \tag{29}$$

*where* $\tilde{f}^{SAM}(X_t) := f(X_t) + \rho \mathbb{E} \|\nabla f_\gamma(X_t)\| + \frac{\rho^2}{2} \mathbb{E} \frac{\nabla f_\gamma^\top \nabla^2 f_\gamma(X_t) \nabla f_\gamma}{\|\nabla f_\gamma\|^2}$

$\Sigma^{SAM}(X_t) := \Sigma^{1,1}(X_t) + \rho(\Sigma^{1,2}(X_t) + \Sigma^{1,2}(X_t)^\top) + \rho^2(\Sigma^{2,2}(X_t) + \frac{1}{2}(\Sigma^{1,3}(X_t) + \Sigma^{1,3}(X_t)^\top))$

$$\Sigma^{1,1}(X_t) := \mathbb{E}\left[\left(\nabla f(X_t) - \nabla f_\gamma(X_t)\right)\left(\nabla f(X_t) - \nabla f_\gamma(X_t)\right)^\top\right]$$

$$\Sigma^{1,2}(X_t) := \mathbb{E}\left[\left(\nabla f(X_t) - \nabla f_\gamma(X_t)\right)\left(\mathbb{E}\left[\frac{\nabla^2 f_\gamma(X_t)\nabla f_\gamma}{\|\nabla f_\gamma\|}\right] - \frac{\nabla^2 f_\gamma(X_t)\nabla f_\gamma}{\|\nabla f_\gamma\|}\right)^\top\right]$$

$$\Sigma^{2,2}(X_t) := \mathbb{E}\left[\left(\mathbb{E}\left[\frac{\nabla^2 f_\gamma(X_t)\nabla f_\gamma}{\|\nabla f_\gamma\|}\right] - \frac{\nabla^2 f_\gamma(X_t)\nabla f_\gamma}{\|\nabla f_\gamma\|}\right)\left(\mathbb{E}\left[\frac{\nabla^2 f_\gamma(X_t)\nabla f_\gamma}{\|\nabla f_\gamma\|}\right] - \frac{\nabla^2 f_\gamma(X_t)\nabla f_\gamma}{\|\nabla f_\gamma\|}\right)^\top\right]$$

$$\Sigma^{1,3}(X_t) := \mathbb{E}\left[\left(\nabla f(X_t) - \nabla f_\gamma(X_t)\right)\left(\mathbb{E}\left[\frac{\nabla^3 f_\gamma(X_t)(\nabla f_\gamma, \nabla f_\gamma)}{\|\nabla f_\gamma\|^2}\right] - \left[\frac{\nabla^3 f_\gamma(X_t)(\nabla f_\gamma, \nabla f_\gamma)}{\|\nabla f_\gamma\|^2}\right]\right)^\top\right]$$

*Additionally, let us take*

$$\rho = \mathcal{O}(\eta^{\frac{1}{3}})$$

*Then, $\{X_t : t \in [0,T]\}$ is an order-1 weak approximation of $\{x_k : k \geq 0\}$, i.e. for each $g \in G$, there exists a constant $C$ independent of $\eta$ such that*

$$\max_{k=0,1,\dots N} |\mathbb{E}g(x_k) - \mathbb{E}g(X_{k\eta})| \leq C\eta$$

*Proof.* We will check the conditions in Lemma B.2. As we apply Lemma B.1, we make the following choices:

$$b(x) = -\nabla \tilde{f}^{SAM}(x)$$
$$\sigma(x) = \Sigma^{SAM}(x)^{\frac{1}{2}}$$

First, for $\forall i = 1, ..., d$, we have

$$\mathbb{E}\Delta_i \overset{LemmaB.1}{=} -\partial_{e_i}\tilde{f}^{SAM}(x)\eta + \mathcal{O}\left(\eta\rho^3\right) \tag{30}$$

$$\mathbb{E}\bar{\Delta}_i \overset{LemmaB.3}{=} -\partial_{e_i}\tilde{f}^{SAM}(x)\eta + \mathcal{O}\left(\eta\rho^3\right) \tag{31}$$

Therefore, we have that for some $K_1(x) \in G$

$$\left|\mathbb{E}\Delta_i - \mathbb{E}\bar{\Delta}_i\right| \leq K_1(x)\eta^2 \tag{32}$$

Second, for $\forall i, j = 1, ..., d$, it holds that

$$\mathbb{E}\Delta_i\Delta_j \overset{LemmaB.1}{=} \partial_{e_i}\tilde{f}^{SAM}(x)\partial_{e_j}\tilde{f}^{SAM}(x)\eta^2 + \Sigma^{SAM}_{(ij)}\eta^2 + \mathcal{O}\left(\eta^2\rho^3\right) \tag{33}$$

$$\mathbb{E}\bar{\Delta}_i\bar{\Delta}_j \overset{LemmaB.3}{=} \partial_{e_i}\tilde{f}^{SAM}(x)\partial_{e_j}\tilde{f}^{SAM}(x)\eta^2 + \Sigma^{SAM}_{(ij)}\eta^2 + \mathcal{O}\left(\eta^2\rho^3\right) \tag{34}$$

Consequently, we have for some $K_2(x) \in G$

$$\left|\mathbb{E}\Delta_i\Delta_j - \mathbb{E}\bar{\Delta}_i\bar{\Delta}_j\right| \leq K_2(x)\eta^2 \tag{35}$$

Third, for $\forall i_j = 1, ..., d$, we have

$$\mathbb{E}\prod_{j=1}^{s}\Delta_{(i_j)} \overset{LemmaB.1}{=} \mathcal{O}\left(\eta^3\right) \tag{36}$$

$$\mathbb{E}\prod_{j=1}^{s}\bar{\Delta}_{(i_j)} \overset{LemmaB.3}{=} \mathcal{O}\left(\eta^3\right) \tag{37}$$

Therefore, for some $K_3(x) \in G$, we have

$$\left|\mathbb{E}\prod_{j=1}^{s}\Delta_{i_j} - \mathbb{E}\prod_{j=1}^{s}\bar{\Delta}_{i_j}\right| \leq K_3(x)\eta^2, \quad \forall s \geq 3 \tag{38}$$

Additionally, for some $K_4(x) \in G$, $\forall i_j = 1, ..., d$,

$$\mathbb{E} \prod_{j=1}^{3} \left| \bar{\Delta}_{i_j} \right| \overset{Lemma B.3}{\leq} K_4(x) \eta^2 \tag{39}$$

By combining the four equations, Eq. 32, Eq. 35, Eq. 38, Eq. 39, we complete the proof. $\qquad\square$

**Proof of Corollary 4.1.1** 1. Under the supposition, the first statement is immediately followed by using the fact:

$$\left| \frac{\rho^2}{2} \mathbb{E} \frac{\nabla f_\gamma^\top \nabla^2 f_\gamma(X_t) \nabla f_\gamma}{\|\nabla f_\gamma\|^2} - \frac{\rho^2}{2} \mathbb{E} \left[ v_1(\nabla^2 f_\gamma(X_t))^\top \nabla^2 f_\gamma(X_t) v_1(\nabla^2 f_\gamma(X_t)) \right] \right| = \mathcal{O}(\rho^3) \tag{40}$$

2. Under the supposition, the second statement is immediately followed by using the fact:

$$\left| \frac{\rho^2}{2} \mathbb{E} \frac{\nabla f_\gamma^\top \nabla^2 f_\gamma(X_t) \nabla f_\gamma}{\|\nabla f_\gamma\|^2} - \frac{\rho^2}{2} \mathbb{E} \left[ v_1(\nabla^2 f_\gamma(X_t))^\top \nabla^2 f_\gamma(X_t) v_1(\nabla^2 f_\gamma(X_t)) \right] \right| = \mathcal{O}(\rho^4) \tag{41}$$

, and

$$\left| \rho \, \mathbb{E} \left[ \frac{\nabla^2 f_\gamma(x) \nabla f_\gamma}{\|\nabla f_\gamma\|} \right] - \rho \, \mathbb{E} \left[ s^* \cdot \nabla^2 f_\gamma(x) \, v_1\left(\nabla^2 f_\gamma(x)\right) \right] \right| = \mathcal{O}(\rho^3), \tag{42}$$

$$\left| \rho \, \mathbb{E} \left[ \frac{\nabla^2 f_\gamma(x) \nabla f_\gamma}{\|\nabla f_\gamma\|} \right] - \rho \, \mathbb{E} \left[ s^* \cdot \lambda_1\left(\nabla^2 f_\gamma(x)\right) v_1\left(\nabla^2 f_\gamma(x)\right) \right] \right| = \mathcal{O}(\rho^3) \tag{43}$$

## C  Proof of Theorem 3.1

**Theorem C.1.** (Generalization Bound) *Assume that the loss function is bounded by L, and the third-order derivative of the loss function is bounded by C. Additionally, we assume $f_\mathcal{D}(x) \leq \mathbb{E}_{\epsilon \sim \mathcal{N}(0, \sigma^2 \mathbb{I}_d)} f_\mathcal{D}(x + \epsilon)$, similar to Foret et al. (2021). For any $\delta \in (0, 1)$ and $\sigma > 0$, with a probability over $1 - \delta$ over the choice of $\mathcal{S} \sim \mathcal{D}^n$, we have*

$$f_\mathcal{D}(x) \leq f_\mathcal{S}(x) + \frac{d\sigma^2}{2} \lambda_1 \left( \nabla^2 f_\mathcal{S}(x) \right) + \frac{C d^3 \sigma^3}{6}$$

$$+ \frac{L}{2\sqrt{n}} \sqrt{d \log \left( 1 + \frac{\|x\|^2}{d\sigma^2} \right) + O(1) + 2 \log \frac{1}{\delta} + 4 \log(n + d)}.$$

*Proof.* We use the PAC-Bayes theory in this proof. In PAC-Bayes theory, $x$ follows a distribution, denoted by $P$, and we express the expected loss over $x$ as follows:

$$f_\mathcal{D}(P) = \mathbb{E}_{x \sim P} \left[ f_\mathcal{D}(x) \right]$$
$$f_\mathcal{S}(P) = \mathbb{E}_{x \sim P} \left[ f_\mathcal{S}(x) \right]$$

For any distribution $P = \mathcal{N}(0, \sigma_P^2 \mathbb{I}_d)$ and $Q = \mathcal{N}(x, \sigma^2 \mathbb{I}_d)$ over $x \in \mathbb{R}^d$, where $P$ is the prior distribution and $Q$ is the posterior distribution, use the general PAC-Bayes theorem in Alquier et al. (2016), for all $\beta > 0$, with a probability at least $1 - \delta$, we have

$$f_\mathcal{D}(Q) \leq f_\mathcal{S}(Q) + \frac{1}{\beta} \left[ \mathsf{KL}(Q \| P) + \log \frac{1}{\delta} + \Psi(\beta, n) \right], \tag{44}$$

where $\Psi$ is defined as

$$\Psi(\beta, n) = \log \mathbb{E}_{x \sim P} \mathbb{E}_\mathcal{S} \left[ \exp \left\{ \beta \left[ f_\mathcal{D}(x) - f_\mathcal{S}(x) \right] \right\} \right].$$

When the loss function is bounded by $L$, then by using the Hoeffding's inequality we have:

$$\Psi(\beta, n) \leq \frac{\beta^2 L^2}{8n}.$$

The task is to minimize the second term of RHS of (44), we thus choose $\beta = \frac{\sqrt{8n\left(\mathsf{KL}(Q\|P)+\log\frac{1}{\delta}\right)}}{L}$.
Then the second term of RHS of (44) is equal to

$$\sqrt{\frac{\mathsf{KL}(Q\|P) + \log\frac{1}{\delta}}{2n}} \times L.$$

The KL divergence between $Q$ and $P$, when they are Gaussian, is given by formula

$$\mathsf{KL}(Q\|P) = \frac{1}{2}\left[\frac{d\sigma^2 + \|x\|^2}{\sigma_P^2} - d + d\log\frac{\sigma_P^2}{\sigma^2}\right].$$

For given posterior distribution $Q$ with fixed $\sigma^2$, to minimize the KL term, the $\sigma_P^2$ should be equal to $\sigma^2 + \|x\|^2/d$. In this case, the KL term is no less than

$$d\log\left(1 + \frac{\|x\|^2}{d\sigma^2}\right).$$

Thus, the second term of RHS is

$$\sqrt{\frac{\mathsf{KL}(Q\|P) + \log\frac{1}{\delta}}{2n}} \times L \geq \sqrt{\frac{d\log\left(1 + \frac{\|x\|^2}{d\sigma^2}\right)}{4n}} \times L \geq L$$

when $\|x\|_2^2 > \sigma^2\left\{\exp(4n/d) - 1\right\}$. Hence, for any $\|x\|^2 > \sigma^2\left\{\exp(4n/d) - 1\right\}$, we have the RHS is greater than the LHS, the inequality is trivial. In the remainder of the proof, we only consider the case:

$$\|x\|^2 < \sigma^2\left(\exp(4n/d) - 1\right). \tag{45}$$

Distribution $P$ is Gaussian centered around 0 with variance $\sigma_P^2 = \sigma^2 + \|x\|^2/d$, which is unknown at the time we set up the inequality, since $x$ is unknown. Meanwhile we have to specify $P$ in advance, since $P$ is the prior distribution. To deal with this problem, we apply the union bound technique (Dziugaite and Roy, 2017; Foret et al., 2021). We set

$$c = \sigma^2\left(1 + \exp(4n/d)\right)$$
$$P_j = \mathcal{N}\left(0, c\exp\left(\frac{1-j}{d}\right)\mathbb{I}_d\right)$$
$$\mathfrak{P} := \left\{P_j : j = 1, 2, \dots\right\}$$

Then the following inequality holds for a particular distribution $P_j$ with probability $1 - \delta_j$ with $\delta_j = \frac{6\delta}{\pi^2 j^2}$

$$f_{\mathcal{D}}(Q) \leq f_{\mathcal{S}}(Q) + \frac{1}{\beta}\left[\mathsf{KL}(Q\|P_j) + \log\frac{1}{\delta_j} + \Psi(\beta, n)\right].$$

Use the well-known equation: $\sum_{j=1}^{\infty}\frac{1}{j^2} = \frac{\pi^2}{6}$, then with probability $1 - \delta$, the above inequality holds with every $j$. We pick

$$j^* := \left\lceil 1 - d\log\frac{\sigma^2 + \|x\|^2/d}{c}\right\rceil = \left\lceil 1 - d\log\frac{\sigma^2 + \|x\|^2/d}{\sigma^2(1 + \exp\{4n/d\})}\right\rceil.$$

Therefore,

$$1 - j^* = \left\lfloor d\log\frac{\sigma^2 + \|x\|^2/d}{c}\right\rfloor$$
$$\Rightarrow \quad \log\frac{\sigma^2 + \|x\|^2/d}{c} \leq \frac{1 - j^*}{d} \leq \log\frac{\sigma^2 + \|x\|^2/d}{c} + \frac{1}{d}$$
$$\Rightarrow \quad \sigma^2 + \|x\|^2/d \leq c\exp\left(\frac{1 - j^*}{d}\right) \leq \exp(1/d)\left[\sigma^2 + \|x\|^2/d\right]$$
$$\Rightarrow \quad \sigma^2 + \|x\|^2/d \leq \sigma_{P_{j^*}}^2 \leq \exp(1/d)\left[\sigma^2 + \|x\|^2/d\right].$$

Thus the KL term could be bounded as follow

$$\mathsf{KL}(Q\|P_{j^*}) = \frac{1}{2}\left[\frac{d\sigma^2 + \|x\|^2}{\sigma_{P_{j^*}}^2} - d + d\log\frac{\sigma_{P_{j^*}}^2}{\sigma^2}\right]$$

$$\leq \frac{1}{2}\left[\frac{d(\sigma^2 + \|x\|^2/d)}{\sigma^2 + \|x\|^2/d} - d + d\log\frac{\exp(1/d)\left(\sigma^2 + \|x\|^2/d\right)}{\sigma^2}\right]$$

$$= \frac{1}{2}\left[d\log\frac{\exp(1/d)\left(\sigma^2 + \|x\|^2/d\right)}{\sigma^2}\right]$$

$$= \frac{1}{2}\left[1 + d\log\left(1 + \frac{\|x\|^2}{d\sigma^2}\right)\right]$$

For the term $\log\frac{1}{\delta_{j^*}}$, with recall that $c = \sigma^2\left(1 + \exp(4n/d)\right)$ and

$j^* = \left\lfloor 1 - d\log\frac{\sigma^2 + \|x\|^2/d}{\sigma^2(1+\exp\{4n/d\})}\right\rfloor$, we have

$$\log\frac{1}{\delta_{j^*}} = \log\frac{(j^*)^2\pi^2}{6\delta} = \log\frac{1}{\delta} + \log\left(\frac{\pi^2}{6}\right) + 2\log(j^*)$$

$$\leq \log\frac{1}{\delta} + \log\frac{\pi^2}{6} + 2\log\left(1 + d\log\frac{\sigma^2\left(1 + \exp(4n/d)\right)}{\sigma^2 + \|x\|^2/d}\right)$$

$$\leq \log\frac{1}{\delta} + \log\frac{\pi^2}{6} + 2\log\left(1 + d\log\left(1 + \exp(4n/d)\right)\right)$$

$$\leq \log\frac{1}{\delta} + \log\frac{\pi^2}{6} + 2\log\left(1 + d\left(1 + \frac{4n}{d}\right)\right)$$

$$\leq \log\frac{1}{\delta} + \log\frac{\pi^2}{6} + \log(1 + d + 4n).$$

Hence, the inequality

$$f_{\mathcal{D}}\left(Q\right) \leq f_{\mathcal{S}}\left(Q\right) + \sqrt{\frac{\mathsf{KL}(Q\|P_{j^*}) + \log\frac{1}{\delta_{j^*}}}{2n}} \times L$$

$$\leq f_{\mathcal{S}}\left(Q\right)$$

$$+ \frac{L}{2\sqrt{n}}\sqrt{1 + d\log\left(1 + \frac{\|x\|^2}{d\sigma}\right) + 2\log\frac{\pi^2}{6\delta} + 4\log(n+d)}$$

$$\leq f_{\mathcal{S}}\left(Q\right)$$

$$+ \frac{L}{2\sqrt{n}}\sqrt{d\log\left(1 + \frac{\|x\|^2}{d\sigma^2}\right) + O(1) + 2\log\frac{1}{\delta} + 4\log(n+d)}.$$

$$\mathbb{E}_{\epsilon\sim\mathcal{N}(0,\sigma^2\mathbb{I}_d)}\left[f_{\mathcal{D}}\left(x+\epsilon\right)\right] \leq \mathbb{E}_{\epsilon\sim\mathcal{N}(0,\sigma^2\mathbb{I}_d)}\left[f_{\mathcal{S}}\left(x+\epsilon\right)\right]$$

$$+ \frac{L}{2\sqrt{n}}\sqrt{d\log\left(1 + \frac{\|x\|^2}{d\sigma^2}\right) + O(1) + 2\log\frac{1}{\delta} + 4\log(n+d)}.$$

Using the assumption that $f_{\mathcal{D}}(x) \leq \mathbb{E}_{\epsilon\sim\mathcal{N}(0,\sigma^2\mathbb{I}_d)}\left[f_{\mathcal{D}}\left(x+\epsilon\right)\right]$, we reach

$$f_{\mathcal{D}}\left(x\right) \leq \mathbb{E}_{\epsilon\sim\mathcal{N}(0,\sigma^2\mathbb{I}_d)}\left[f_{\mathcal{S}}\left(x+\epsilon\right)\right]$$

$$+ \frac{L}{2\sqrt{n}}\sqrt{d\log\left(1 + \frac{\|x\|^2}{d\sigma^2}\right) + O(1) + 2\log\frac{1}{\delta} + 4\log(n+d)}.$$

Using the second-order Taylor expansion for $f_{\mathcal{S}}(x + \epsilon)$, we obtain

$$f_{\mathcal{S}}(x + \epsilon) = f_{\mathcal{S}}(x) + \epsilon^T \nabla_x f_{\mathcal{S}}(x) + \frac{1}{2}\epsilon^T \nabla^2 f_{\mathcal{S}}(x)\epsilon + \frac{1}{6}\sum_{i_1,i_2,i_3}\frac{\partial^3 f_{\mathcal{S}}(x + t\epsilon)}{\partial x_{i_1}\partial x_{i_2}\partial x_{i_3}}\epsilon_{i_1}\epsilon_{i_2}\epsilon_{i_3}$$

$$\leq f_{\mathcal{S}}(x) + \epsilon^T \nabla_x f_{\mathcal{S}}(x) + \frac{1}{2}\lambda_1\left(\nabla^2 f_{\mathcal{S}}(x)\right)\|\epsilon\|_2^2 + \frac{1}{6}\sum_{i_1,i_2,i_3}\frac{\partial^3 f_{\mathcal{S}}(x + t\epsilon)}{\partial x_{i_1}\partial x_{i_2}\partial x_{i_3}}\epsilon_{i_1}\epsilon_{i_2}\epsilon_{i_3},$$

where $t \in [0, 1]$. Thanks to $\mathbb{E}_{\epsilon \sim \mathcal{N}(0,\sigma^2\mathbb{I}_d)}\left[\|\epsilon^2\right] \leq \mathbb{E}_{\epsilon \sim \mathcal{N}(\mathbf{0},\mathbb{I}_d)}\left[\|\epsilon^2\right] = d\sigma^2$, we have

$$\mathbb{E}_{\epsilon \sim \mathcal{N}(0,\sigma^2\mathbb{I}_d)}\left[f_{\mathcal{S}}\left(x + \epsilon\right)\right] \leq f_{\mathcal{S}}\left(x\right) + \frac{d\sigma^2}{2}\lambda_1\left(\nabla^2 f_{\mathcal{S}}(x)\right)$$
$$+ \frac{Cd^3}{6}\mathbb{E}_{\epsilon_1 \sim \mathcal{N}(0,\sigma^2)}\left[|\epsilon_1|\right]\mathbb{E}_{\epsilon_2 \sim \mathcal{N}(0,\sigma^2)}\left[|\epsilon_2|\right]\mathbb{E}_{\epsilon_3 \sim \mathcal{N}(0,\sigma^2)}\left[|\epsilon_3|\right]$$
$$\leq f_{\mathcal{S}}\left(x\right) + \frac{d\sigma^2}{2}\lambda_1\left(\nabla^2 f_{\mathcal{S}}(x)\right) + \frac{Cd^3}{6}\left(\mathbb{E}_{\epsilon_1 \sim \mathcal{N}(0,\sigma^2)}\left[\epsilon_1^2\right]^{1/2}\right)^3$$
$$= f_{\mathcal{S}}\left(x\right) + \frac{d\sigma^2}{2}\lambda_1\left(\nabla^2 f_{\mathcal{S}}(x)\right) + \frac{Cd^3\sigma^3}{6}.$$

By the assumption $f_{\mathcal{D}}(x) \leq \mathbb{E}_{\epsilon \sim \mathcal{N}(0,\sigma^2\mathbb{I}_d)}f_{\mathcal{D}}(x + \epsilon)$, we reach

$$f_{\mathcal{D}}(x) \leq f_{\mathcal{S}}\left(x\right) + \frac{d\sigma^2}{2}\lambda_1\left(\nabla^2 f_{\mathcal{S}}(x)\right) + \frac{Cd^3\sigma^3}{6}$$
$$+ \frac{L}{2\sqrt{n}}\sqrt{d\log\left(1 + \frac{\|x\|^2}{d\sigma^2}\right) + O(1) + 2\log\frac{1}{\delta} + 4\log(n + d)}.$$

$\square$

## D  Theoretical Properties of Eigen-SAM

In this subsection, first we show that Eq. 8 can indeed improve the alignment for moderate $\alpha$. Let $\omega := \cos(\frac{\nabla f_\gamma(x)}{\|\nabla f_\gamma(x)\|}, v)$, without loss of generalization, we suppose $\omega > 0$. We only consider the case $\omega > \frac{\sqrt{2}}{2}$, since in the case $0 < \omega \leq \frac{\sqrt{2}}{2}$, any $\alpha > 0$ will enhance the alignment. Using fundamental mathematics to solve the inequality, we obtain

**Proposition D.1.** *Let $\omega > \frac{\sqrt{2}}{2}$, for any $\alpha \in \left(0, \frac{2\omega\sqrt{1-\omega^2}}{2\omega^2-1}\right)$, we have $\cos(\frac{\nabla f_\gamma(x)}{\|\nabla f_\gamma(x)\|} + \alpha v_\perp, v) > \omega$.*

Proposition D.1 shows our update can indeed improve the alignment for a wide range of $\alpha$. For example, if $\cos(\nabla f_\gamma(x), v) = 0.8$, then $\alpha$ can be any value in $(0, 3.43)$; if $\cos(\nabla f_\gamma(x), v) = 0.9$, then $\alpha$ can be any value in $(0, 1.27)$.

Next, inspired by Si and Yun (2024), we have the following convergence rate for stochastic Eigen-SAM on non-convex function:

**Theorem D.2.** (Convergence rate) *Consider a $\beta$-smooth function $f$ satisfying $f^* = \inf_x f(x) > -\infty$, let $\Delta := f(x_0) - f^*$, and assume the mini-batch variance is bounded by $\sigma^2$. Under Eigen-SAM, starting at $x_0$ with any perturbation size $\rho > 0$ and step size $\eta = \min\{\frac{1}{2\beta}, \frac{\sqrt{\Delta}}{\sqrt{\beta\sigma^2 T}}\}$ to minimize $f$, we have*

$$\frac{1}{T}\sum_{t=0}^{T-1}\mathbb{E}\|\nabla f_\gamma(x_t)\|^2 \leq \mathcal{O}\left(\frac{\beta\Delta}{T} + \frac{\sqrt{\beta\sigma^2\Delta}}{\sqrt{T}}\right) + \beta^2(\rho^2 + \alpha^2)$$

*Proof.* By the definition of $\beta$-smoothness, we have

$$\mathbb{E}f(x_{t+1}) \leq \mathbb{E}f(x_t) - \eta\mathbb{E}\langle\nabla f(x_t), \nabla f(x_t + \rho\epsilon)\rangle + \frac{\beta\eta^2}{2}\mathbb{E}\|\nabla f(x_t + \rho\epsilon)\|^2$$

$$\leq \mathbb{E}f(x_t) - \eta\mathbb{E}\langle\nabla f(x_t), \nabla f(x_t + \rho\epsilon)\rangle$$
$$+ \beta\eta^2\left(\mathbb{E}\|\nabla f(x_t + \rho\epsilon)\|^2 + \mathbb{E}\|\nabla f(x_t + \rho\epsilon) - \nabla f(x_t)\|^2\right)$$

$$\leq \mathbb{E}f(x_t) - \eta\mathbb{E}\langle\nabla f(x_t), \nabla f(x_t + \rho\epsilon)\rangle + \beta\eta^2\left(\mathbb{E}\|\nabla f(x_t + \rho\epsilon)\|^2 + \sigma^2\right)$$

$$= \mathbb{E}f(x_t) - \frac{\eta}{2}\mathbb{E}\|\nabla f(x_t)\|^2 - \frac{\eta}{2}\mathbb{E}\|\nabla f(x_t + \rho\epsilon)\|^2$$
$$+ \frac{\eta}{2}\mathbb{E}\|\nabla f(x_t) - \nabla f(x_t + \rho\epsilon)\|^2 + \beta\eta^2\left(\mathbb{E}\|\nabla f(x_t + \rho\epsilon)\|^2 + \sigma^2\right)$$

$$\leq \mathbb{E}f(x_t) - \frac{\eta}{2}\mathbb{E}\|\nabla f(x_t)\|^2 + \frac{\beta^2\eta}{2}\mathbb{E}\|x_t - (x_t + \rho\epsilon)\|^2 + \beta\sigma^2\eta^2$$

$$= \mathbb{E}f(x_t) - \frac{\eta}{2}\mathbb{E}\|\nabla f(x_t)\|^2 + \frac{\beta^2\eta(\rho^2 + \alpha^2)}{2} + \beta\sigma^2\eta^2.$$

Rearranging this inequality and deviding both sides by $\frac{\eta T}{2}$, we have

$$\frac{1}{T}\sum_{t=0}^{T-1}\mathbb{E}\|\nabla f(x_t)\|^2 \leq \frac{2}{\eta T}\left(\mathbb{E}f(x_0) - \mathbb{E}f(x_T)\right) + \beta^2(\rho^2 + \alpha^2) + 2\beta\sigma^2\eta$$

$$\leq \frac{2\Delta}{\eta T} + \beta^2(\rho^2 + \alpha^2) + 2\beta\sigma^2\eta.$$

By using $\eta = \min\{\frac{1}{2\beta}, \frac{\sqrt{\Delta}}{\sqrt{\beta\sigma^2 T}}\}$, we complete the proof. $\square$

# E  Additional Experimental Details

For hyperparameter $\rho$, we follow the guidelines by Foret et al. (2021), setting $\rho$ to 0.05 for CIFAR-10 and Fashion-MNIST, 0.01 for SVHN, and 0.1 for CIFAR-100. We ensure that SAM and Eigen-SAM use the same $\rho$ for a fair comparison. Additionally, we tune the hyperparameter $\alpha$ for Eigen-SAM over {0.05, 0.1, 0.2} using 10% of the training set as a validation set. We find that $\alpha = 0.2$ works the best for almost all cases. Therefore, we report the performance with $\alpha = 0.2$ for all experiments to demonstrate that Eigen-SAM does not require extensive hyperparameter tuning. We run three independent repeat experiments with different weight initializations and data shuffling. Because SAM and Eigen-SAM require twice the runtime, we allow SGD to train for twice the number of epochs. All our experiments were conducted on NVIDIA RTX 4090 24GB GPUs.

# F  Additional experiment results

Table 3: Test accuracy on CIFAR-10 for different values of p (interval steps for estimating eigenvectors) using Eigen-SAM.

| Architechture | p=100 | p=200 | p=500 | p=1000 | SAM |
|---|---|---|---|---|---|
| ResNet18 | $95.9_{\pm0.2}$ | $95.9_{\pm0.1}$ | $95.7_{\pm0.2}$ | $95.8_{\pm0.1}$ | $95.5_{\pm0.1}$ |
| WideResNet-28-10 | $96.8_{\pm0.1}$ | $96.7_{\pm0.1}$ | $96.8_{\pm0.1}$ | $96.7_{\pm0.1}$ | $96.5_{\pm0.1}$ |

Table 4: Test accuracy on CIFAR-100 for different values of p (interval steps for estimating eigenvectors) using Eigen-SAM.

| Architechture | p=100 | p=200 | p=500 | p=1000 | SAM |
|---|---|---|---|---|---|
| ResNet18 | $78.3_{\pm 0.2}$ | $78.2_{\pm 0.2}$ | $78.2_{\pm 0.2}$ | $78.3_{\pm 0.1}$ | $77.4_{\pm 0.2}$ |
| WideResNet-28-10 | $82.8_{\pm 0.1}$ | $82.7_{\pm 0.2}$ | $82.7_{\pm 0.1}$ | $82.6_{\pm 0.1}$ | $82.0_{\pm 0.2}$ |

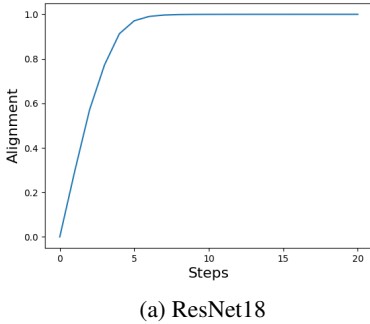
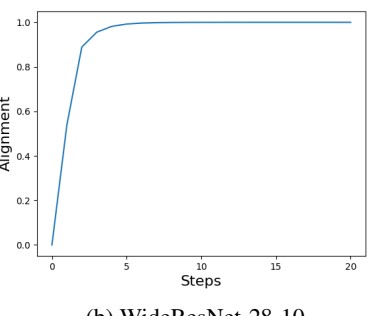

(a) ResNet18          (b) WideResNet-28-10

Figure 5: The effect of the number of Hessian-vector product steps in Algorithm 1 (power iteration) on the alignment of the estimated vector with the top eigenvalue. The dataset is CIFAR-100, and the models are ResNet18 and WideResNet-28-10 at mid-training stage (100th epoch).

## G    Limitaions

A limitation of this work is the additional memory and time required to estimate the top eigenvalue of the Hessian matrix. Improving the efficiency of Eigen-SAM is a direction for future research.

