# OpenReview forum: "Explicit Eigenvalue Regularization Improves Sharpness-Aware Minimization"
_NeurIPS.cc/2024/Conference — NeurIPS 2024 poster_

### Official Review · Reviewer_B3hY · 2024-06-30

**Soundness:** 3
**Presentation:** 4
**Contribution:** 3
**Rating:** 4
**Confidence:** 3

**Summary:**

This paper proposes a novel method, Eigen-SAM, as an improvement to Sharpness-Aware Minimization (SAM). The paper theoretically elucidates the relationship between the top eigenvalue of the Hessian matrix and generalization error and models the dynamics of SAM using a third-order stochastic differential equation (SDE). Eigen-SAM aims to more effectively minimize sharpness by intermittently estimating the top eigenvector of the Hessian matrix. The effectiveness of Eigen-SAM is validated through experiments across multiple small datasets.

**Strengths:**

Soundness
- The technical claims, experimental methods, and research methodology are robust, and the central claims are well-supported by evidence.
- The necessity of the third-order SDE is well-explained in section 4.

Presentation
- The presentation is generally clear, providing context by comparing with existing research.

Contribution
- This study contributes to the field by deepening the theoretical understanding of Sharpness-Aware Minimization and proposing a new algorithm that enhances practical performance.

Originality
- Novelty by proposing Eigen-SAM as an improvement to SAM, theoretically demonstrating the relationship between the top eigenvalue of the Hessian matrix and generalization error through a third-order SDE.

**Weaknesses:**

- Experiments are limited to ResNet models, raising questions about the method's effectiveness with other models. Comparative experiments with ViT and other architectures are strongly recommended.
- The lack of publicly available implementation may pose a barrier to community adoption.
- Demonstrating ablation studies on parameters such as k (interval for power method) and p (iterations for power method) would make the method more useful for the community.
- Showing effectiveness in more practical settings like ImageNet would impact my score.
- Comparisons with other optimization methods using eigenvalues would strengthen this paper.

**Questions:**

- More specific guidelines on the frequency of top eigenvector estimation and parameter selection in Eigen-SAM implementation would be helpful. How are these parameters selected?
- Is Figure 4(a) showing sensitivity analysis for alpha rather than rho? This is a bit confusing to me.
- Does the fact that v updates only once every 100 steps in equation 8 imply stability in the inner product of the gradient with the direction of the maximum eigenvalue? Could you provide an intuitive explanation?
- Are parameters other than alpha being tuned? If 90% of the training set is used as training data and the remaining 10% as validation data for Eigen-SAM, and no validation data is used for other optimizers, is my understanding correct?
- Wen et al. (2022) interpret SAM learning with full batch training as regularizing the maximum eigenvalue. Could increasing the batch size improve alignment for SAM? Additionally, how does batch size affect alignment, maximum eigenvalue, and test accuracy in Eigen-SAM?
- For Table 1, can you also show the results of how much the maximum eigenvalue improved?

**Limitations:**

- The authors acknowledge the increased computational cost of implementing Eigen-SAM, which may limit its feasibility in some applications. Future research should explore methods to reduce this computational cost and develop more efficient implementations. It would also be beneficial to consider cases where the applicability of the proposed method may be limited.
- A significant limitation is that experiments are conducted only on ResNet models, and the effectiveness on more modern models like ViT is unknown.

---

> ### Author Rebuttal · Authors · 2024-08-07
>
> Thank you for your insightful comments and constructive suggestions. Below we address your comments:
>
> Weaknesses:
>
> * **Experiments are limited to ResNet models, raising questions about the method's effectiveness with other models. Comparative experiments with ViT and other architectures are strongly recommended.** In Table 1 of the rebuttal PDF, we additionally show the results of fine-tuning on CIFAR-10/CIFAR-100 using ViT-B-16 pretrained on ImageNet, showing that Eigen-SAM consistently outperforms original SAM.
>
> * **The lack of publicly available implementation may pose a barrier to community adoption.** We are committed to supporting the research community and plan to open-source our code upon acceptance of the paper.
>
> * **Demonstrating ablation studies on parameters such as k (interval for power method) and p (iterations for power method) would make the method more useful for the community.** In Table 2 and Table 3 of our rebuttal PDF, we show how larger values of k affect generalization performance and observe that even setting k to 1000 (resulting in less than 1% additional overhead) can retain most of the performance gains. In Figure 1 of the rebuttal PDF, we demonstrate the efficient convergence speed of Algorithm 1, which requires minimal adjustment to p.
>
> * **Showing effectiveness in more practical settings like ImageNet would impact my score.** Due to computational costs and time constraints, we were unable to complete experiments on ImageNet during the rebuttal period, but we are very willing to include it in our future plans.
>
> * **Comparisons with other optimization methods using eigenvalues would strengthen this paper.** To the best of our knowledge, we are the first to use eigenvalue regularization in a SAM variant.
>
> Questions:
>
> * **More specific guidelines on the frequency of top eigenvector estimation and parameter selection in Eigen-SAM implementation would be helpful. How are these parameters selected?** Among our hyperparameters, alpha is tuned on the validation set. There is no consistent selection principle for k and p. A larger k improves the quality of the estimated top eigenvector during training but incurs higher computational costs, requiring a trade-off between the two. For more experiments on hyperparameter selection, please refer to Tables 2 and 3 as well as Figure 1 in our rebuttal PDF.
>
> * **Is Figure 4(a) showing sensitivity analysis for alpha rather than rho? This is a bit confusing to me.** This is a typo, and we apologize for the confusion it caused. We will correct it in future versions.
>
> * **Does the fact that v updates only once every 100 steps in equation 8 imply stability in the inner product of the gradient with the direction of the maximum eigenvalue? Could you provide an intuitive explanation?** Yes, it means that the decay rate of the alignment between the estimated top eigenvector and the real top eigenvector is relatively slow. We refer to Figure 2 in our rebuttal PDF for empirical evidence of this phenomenon.
>
> * **Are parameters other than alpha being tuned? If 90% of the training set is used as training data and the remaining 10% as validation data for Eigen-SAM, and no validation data is used for other optimizers, is my understanding correct?** For other hyper-parameters, such as learning rate and perturbation radius, we followed previous work and kept them consistent across all optimizers.
>
> * **Wen et al. (2022) interpret SAM learning with full batch training as regularizing the maximum eigenvalue. Could increasing the batch size improve alignment for SAM? Additionally, how does batch size affect alignment, maximum eigenvalue, and test accuracy in Eigen-SAM?** Previous work [1] has empirically shown that increasing the batch size can actually increase the sharpness of the model, thereby harming generalization performance, highlighting the gap between the theory and practice of SAM. Our theory holds for any batch size, unlike [2], which is only applicable to specific batch sizes. Therefore, we do not have theoretical results related to batch size and have not conducted batch size related experiments.
>
> * **For Table 1, can you also show the results of how much the maximum eigenvalue improved?** Yes, here is the table of the top eigenvalue:
> |Model| Method | CIFAR-10 | CIFAR-100 |
> |----------|----------|----------|----------|
> |ResNet-18|SAM | 5.0 | 5.7 |
> ||Eigen-SAM | 4.2 | 4.9 |
> |ResNet-50|SAM | 4.2 | 4.3 |
> ||Eigen-SAM | 3.8 | 3.7 |
> |WideResNet-28-10|SAM | 1.2 | 1.7 |
> ||Eigen-SAM | 1.0 | 1.4 |
>
> References
>
> [1] Andriushchenko, M., & Flammarion, N. (2022, June). Towards understanding sharpness-aware minimization. In International Conference on Machine Learning (pp. 639-668). PMLR.
>
> [2] Wen, K., Ma, T., & Li, Z. (2023). How Sharpness-Aware Minimization Minimizes Sharpness?. In The Eleventh International Conference on Learning Representations.

---

> > ### Comment · Reviewer_B3hY · 2024-08-12
> > **Official Comment by Reviewer B3hY**
> >
> > Thanks for the response. Regarding the suggestion to compare with other optimization methods using eigenvalues, there are SAM variants that constrain the maximum eigenvalue, such as those proposed in [1] (Hessian Trace, which regularize eigenvalue indirectly) and [2] (Eigenvalue of FIM). Although these methods are derived differently, evaluating the performance of similar approaches that aim to improve SAM is important for the community. Additionally, since the current experimental results are limited to a narrow set of models, there are concerns about the broader effectiveness of the proposed method. I will maintain my scores.
> >
> > [1] CR-SAM: Curvature Regularized Sharpness-Aware Minimization
> >
> > [2] Fisher SAM: Information Geometry and Sharpness Aware Minimisation

---

> ### Author Response · Authors · 2024-08-12
>
> Thank you for your thoughtful feedback and for bringing up the comparison with CR-SAM and Fisher SAM. However, while CR-SAM aims to regularize the trace of the Hessian, in high-dimensional settings, the trace often differs significantly from the top eigenvalue, and Fisher SAM does not work on eigenvalues because it aims to find the perturbation w.r.t. the geometry specified by the Fisher distance. Therefore, it is difficult to compare these methods beyond final performance. Nevertheless, we are willing to discuss CR-SAM and Fisher SAM in related work.
>
> Due to limited computational resources, we acknowledge that we were unable to expand the scale of our experiments as much as possible. However, the theoretical developments/derivations are the main contributions of our work. We hope that you can focus more on the theoretical aspects, as better understanding the working principles of SAM can often guide the design of more principled algorithms, which is also beneficial to the community.

---

> ### Author Response · Authors · 2024-08-13
>
> Dear Reviewer B3hY,
>
> We believe that we have addressed most of your questions and concerns, except for conducting additional experiments on ImageNet due to our limited computational resources. We hope you will give greater consideration to the theoretical contributions of our paper, which establish a connection between generalization loss and the top eigenvalue of the Hessian matrix. Additionally, we have developed a third-order SDE of SAM to study its dynamics and identify the factors important for minimizing the top eigenvalue.
>
> We would like to note that our theories remarkably advance two important theoretical works in SAM [1] and [2]. Specifically, comparing to [1], firstly, their theory requires a much longer training time $Θ(η^{−1}ρ^{−2})$, compared to our $Θ(η^{−1})$. Therefore, our SDE corresponds to one phase one of their main analysis in terms of the time scale, during which they do not draw any conclusions about implicit bias. In contrast, our SDE (6), which indicates the implicit bias consists of three components with different scales, is much richer in this phase. Secondly, they require $ηln(1/ρ)$ be sufficiently small, leading ρ to be exponentially smaller than η, whereas our theory works for a more practical range, $ρ = O(η^{1/3}$). Comparing to [2], our theorem has the following two advantages: firstly, we allows ρ to take larger values ($η^{1/3}$ compared to $η^{1/2}$ in [2]), which is more consistent with real-world settings, or equivalently, our SDE has a lower approximation error for fixed ρ; secondly, our SDE reveals SAM’s implicit bias on the Hessian matrix, which occurs as the gradient of the Hessian along the direction of the gradient.
>
>
>
> [1] Wen, K., Ma, T., and Li, Z. (2022). How does sharpness-aware minimization minimize sharpness? CoRR, abs/2211.05729.
>
> [2] Compagnoni, E. M., Biggio, L., Orvieto, A., Proske, F. N., Kersting, H., and Lucchi, A. (2023). An sde for modeling sam: Theory and insights. In International Conference on Machine Learning, pages 25209–25253. PMLR.
>
> Please let us know if there is anything else we can do to increase your recognition of our work.

---

> ### Comment · Reviewer_B3hY · 2024-08-14
> **Official Comment by Reviewer B3hY**
>
> I would like to express my sincere appreciation for detailed response and the considerable effort authors have invested in this research. I acknowledge authors' theoretical contributions are commendable.
>
> However, my primary concern remains with the empirical evaluation. I maintain my current score as I believe that the empirical results presented, while promising, do not meet the soundness required for acceptance at a conference of NeurIPS's caliber. Specifically, the limited experimental setup and the observed marginal improvements, though noteworthy, may not sufficiently demonstrate the broader applicability and effectiveness of the proposed approach.
>
> I recommend further validation of this work on more practical datasets and across multiple model architectures. Such additional evaluations would strengthen authors' findings and potentially make authors work more suitable for submission to future venues.

---

> ### Author Response · Authors · 2024-08-14
>
> Dear Reviewer B3hY,
>
> We appreciate your feedback on our work and thank you for acknowledging our developed theories.
>
> While we agree with your point that further validation of this work on more practical datasets and across multiple model architectures could strengthen our findings, we believe that the number of experiments across multiple model architectures (ResNet, WideResNet-28-10, VIT-B-16: please refer to **the pdf of the global response**) we have conducted in our paper, along with the additional ones in the rebuttal, is appropriate for a theory-based paper. In your comparison with [1, 2], it’s worth noting that [1] did not include any experiments, while [2] performed some experiments to demonstrate certain behaviors. We kindly suggest that the reviewer take this aspect into consideration.
>
> [1] Wen, K., Ma, T., and Li, Z. (2022). How does sharpness-aware minimization minimize sharpness? CoRR, abs/2211.05729.
>
> [2] Compagnoni, E. M., Biggio, L., Orvieto, A., Proske, F. N., Kersting, H., and Lucchi, A. (2023). An sde for modeling sam: Theory and insights. In International Conference on Machine Learning, pages 25209–25253. PMLR.
>
> Best regards,

---

> > ### Comment · Reviewer_B3hY · 2024-08-14
> > **Official Comment by Reviewer B3hY**
> >
> > Thank you for your response and I aknowledge for the additional experiments you provided.
> >
> > However, I must reiterate my earlier point regarding the scope of your experiments. The two related works[1,2] I mentioned that improve SAM have demonstrated their methods on ImageNet scale datasets. In contrast, the studies you referenced focus more on theoretical understanding or exploring phenomena from different perspectives, rather than proposing new methods. While you have proposed Eigen-SAM and provided evidence of its effectiveness, the limited experimental result, specifically,  CIFAR scale experiments with ResNet-based and ViT-based architecture do not, in my view, sufficiently establish its efficacy.
> >
> > I encourage you to consider this aspect in further revisions or future work.
> >
> > [1] CR-SAM: Curvature Regularized Sharpness-Aware Minimization
> >
> > [2] Fisher SAM: Information Geometry and Sharpness Aware Minimisation

---

> > > ### Author Response · Authors · 2024-08-14
> > >
> > > Dear Reviewer B3hY,
> > >
> > > Thank you for responding to our message and giving us the opportunity to share our thoughts. While we appreciate the contributions of CR-SAM and Fisher SAM and are willing to discuss these works in the related work section, we believe that the theoretical contributions of these papers and ours are different.
> > >
> > > CR-SAM aims to minimize the trace of the Hessian matrix, whereas Fisher SAM seeks to find adversarial models within a ball around $\theta$ with respect to the Fisher distance. Moreover, the theories in these papers primarily involve the extension of PAC-Bayes in SAM for their specific cases. As mentioned before, one of the main contributions of our paper is to advance the theories of [1, 2], which are two significant theoretical works in sharpness-aware minimization, i.e., ours offers a better way to study the dynamics of SAM and the convergence for more realistic $\rho$.
> > >
> > > We further build on these theories to propose a practical method that demonstrates our claims: (i) using third-order expansion, we can approximate the dynamics of SAM more accurately, and (ii) our approach can obtain smaller top eigenvalues, leading to improved performance comparing to SAM.
> > >
> > > We believe that it is not appropriate to use the experimental standards of CR-SAM/Fisher SAM for our case, especially since experiments on ImageNet are not affordable for many due to resource limitations. Moreover, in addition to [1, 2], there are many published works in the SAM domain that did not conduct experiments on ImageNet.
> > >
> > > [1] Wen, K., Ma, T., and Li, Z. (2022). How does sharpness-aware minimization minimize sharpness? CoRR, abs/2211.05729.
> > >
> > > [2] Compagnoni, E. M., Biggio, L., Orvieto, A., Proske, F. N., Kersting, H., and Lucchi, A. (2023). An sde for modeling sam: Theory and insights. In International Conference on Machine Learning, pages 25209–25253. PMLR.

---

### Official Review · Reviewer_zC7c · 2024-07-05

**Soundness:** 3
**Presentation:** 2
**Contribution:** 2
**Rating:** 6
**Confidence:** 3

**Summary:**

This study establishes a theoretical connection between the top eigenvalue of the Hessian matrix and generalization error through an extended PAC-Bayes theorem. Highlighting the importance of perturbation-eigenvector alignment in mitigating sharpness, this study introduces Eigen-SAM. This method intermittently estimates the top eigenvector to enhance alignment, leading to enhanced sharpness minimization. Experimental findings demonstrate improvements in both test accuracy and robustness when compared with conventional SAM and SGD methods.

**Strengths:**

This research introduces theoretical advancements aimed at enhancing Sharpness-Aware Minimization (SAM) and proposes an algorithm called Eigen-SAM based on these theoretical foundations. Additionally, supportive experimental evidence is presented in this study.

**Weaknesses:**

The experimental findings exhibit limited strength, indicating the necessity for additional analysis and further experimentation:

A. The experimental results do not sufficiently demonstrate the superiority of Eigen-SAM over SAM.

B. The experiments were conducted using smaller scale models and datasets, where concerns regarding generalization are relatively minimal.

C. Additionally, a comprehensive analysis of computational costs is required.

**Questions:**

A. Could you offer theoretical evidence elucidating the poor alignment in Section 5.1?

B. What is the computational cost in comparison to other algorithms? Can Eigen-SAM be feasibly implemented in large-scale models?

**Limitations:**

Please refer to the Weaknesses.

---

> ### Author Rebuttal · Authors · 2024-08-07
>
> Thank you for your detailed and insightful feedback. We address your concerns below:
>
> * **The experimental findings exhibit limited strength, indicating the necessity for additional analysis and further experimentation: A. The experimental results do not sufficiently demonstrate the superiority of Eigen-SAM over SAM. B. The experiments were conducted using smaller scale models and datasets, where concerns regarding generalization are relatively minimal.** Our main goal is to expand the existing theories on understanding sharpness-aware minimization (SAM) and to bridge the substantial gap between current theory and practice, rather than to propose a highly practical, and very scalable optimizer to replace SAM. Therefore, we did not conduct experiments on large datasets, such as ImageNet. Nevertheless, in Table 1 of the rebuttal PDF, we show the results of fine-tuning on CIFAR-10/CIFAR-100 using ViT-B-16 pretrained on ImageNet, showing that Eigen-SAM consistently outperforms original SAM.
>
> * **C. Additionally, a comprehensive analysis of computational costs is required. Questions B. What is the computational cost in comparison to other algorithms? Can Eigen-SAM be feasibly implemented in large-scale models?** Our additional overhead comes from running the Hessian-vector product p times every k steps to estimate the top eigenvector. The Hessian-vector product roughly takes 1-2 times the time required to compute the gradient, so the overhead of our algorithm is approximately $(2+p/k)$ to $(2+2p/k)$ times that of SGD, compared to 2 times for standard SAM. For larger models, the computation time for the Hessian-vector product hardly increases. We refer to [1] for a more detailed analysis of the computation cost of Hessian-vector products. In Table 2 and Table 3 of our rebuttal PDF, we show how larger values of k affect generalization performance and observe that even setting k to 1000 (resulting in less than 1% additional overhead) can retain most of the performance gains. Regarding the question of whether larger models require more frequent estimation of the top eigenvector, we show in Figure 2 of our rebuttal PDF the rate of decay in the quality of the estimated top eigenvector. For larger models, this decay rate only slightly increases and generally decays relatively slowly.
>
> * **Questions: A. Could you offer theoretical evidence elucidating the poor alignment in Section 5.1?** We are sorry that we cannot provide a theoretical proof for this phenomenon. This precisely highlights the gap between the theory and practice of SAM.
>
> References
>
> [1] Dagréou, et al., "How to compute Hessian-vector products?", ICLR Blogposts, 2024.

---

> > ### Comment · Reviewer_zC7c · 2024-08-10
> >
> > Thank you for conducting the additional experiments and providing the discussions in response to the review. The authors' rebuttal has addressed most of the previously raised concerns, and I am inclined to give a positive review score. I suggest that the experimental results be included in the manuscript.

---

> ### Author Response · Authors · 2024-08-12
>
> Thank you for your positive feedback and for increasing the score. We will include the experimental results in the manuscript as you suggested.

---

### Official Review · Reviewer_WYKu · 2024-07-12

**Soundness:** 3
**Presentation:** 2
**Contribution:** 3
**Rating:** 5
**Confidence:** 3

**Summary:**

The paper focuses on improving Sharpness-Aware Minimization (SAM) by introducing a novel approach called Eigen-SAM. It establishes a theoretical connection between the top eigenvalue of the Hessian matrix and generalization error using an extended PAC-Bayes theorem. The authors derive a third-order stochastic differential equation (SDE) to model the dynamics of SAM, revealing lower approximation error compared to second-order SDEs. The paper emphasizes the importance of perturbation-eigenvector alignment in reducing sharpness. Eigen-SAM intermittently estimates the top eigenvector to enhance this alignment, leading to better sharpness minimization. Extensive experiments demonstrate the effectiveness of Eigen-SAM in improving test accuracy and robustness over standard SAM and SGD across multiple datasets and model architectures.

**Strengths:**

1. The paper provides a strong theoretical foundation by connecting the top eigenvalue of the Hessian to generalization error and developing a third-order SDE for SAM.

2. The paper introduces Eigen-SAM, which improves upon standard SAM by focusing on perturbation-eigenvector alignment.

3. Experiments across datasets and models validate the theoretical insights and demonstrate the practical benefits of Eigen-SAM.

**Weaknesses:**

1. The paper claims that it's the first work establishing the relationship between the top eigenvalue and generalization error. However, some works [1,2] have discussed related topics.

2. The generalization bound given in Theorem 4.1, based on the result of the original SAM paper, considers any random perturbation rather than worst-case perturbation as SAM does in practice. Hence, the theoretical conclusion might be less instructive.

3. Empirical results are inadequate. Test on large-scale datasets like ImageNet-1K is common in SAM-related works. That would be better if there were more experiments on other tasks like noisy label task.

4. Eigen-SAM requires extra computational cost for estimating the top eigenvector. While the authors claim that it only needs 5%-10% overhead, is the overhead related to the size of model architectures? Whether the estimation need more frequent if tested on large-scale datasets?

**Minor:**

- Typo in line 249: should be $v$
- Typo in line 290: should be $\rho$

[1] Wen, Kaiyue, Tengyu Ma, and Zhiyuan Li. "How does sharpness-aware minimization minimize sharpness?." arXiv preprint arXiv:2211.05729 (2022).

[2] Kaur, Simran, Jeremy Cohen, and Zachary Chase Lipton. "On the maximum hessian eigenvalue and generalization." Proceedings on. PMLR, 2023.

**Questions:**

1. Does third-order SDE shown in Figure 2 & 3 estimate the top eigenvector on each mini-batch or once every $k$ mini-batch?

**Limitations:**

The authors have mentioned in Section A.5 that Eigen-SAM requires additional computational time.

---

> ### Author Rebuttal · Authors · 2024-08-07
>
> Thanks for your helpful comments and insightful suggestions. We address your questions one by one below:
>
> Weaknesses:
>
> **1. The paper claims that it's the first work establishing the relationship between the top eigenvalue and generalization error. However, some works [1,2] have discussed related topics.** We would like to clarify that, to the best of our knowledge, we are the first to establish theoretical generalization bounds based on the top eigenvalue. The paper [1] you mentioned analyzes the dynamics and implicit bias of the SAM algorithm but does not establish a connection with generalization. The paper [2] you mentioned is purely experimental work, which is significantly different from our theoretical bound.
>
> **2. The generalization bound given in Theorem 4.1, based on the result of the original SAM paper, considers any random perturbation rather than worst-case perturbation as SAM does in practice. Hence, the theoretical conclusion might be less instructive.** Our Theorem 4.1 aims to establish a relationship between the top eigenvalue of the Hessian matrix and generalization. We acknowledge that it may not be practically instructive. In practice, we rely on our proposed algorithm, Eigen-SAM, to regularize the largest eigenvalue rather than directly optimizing the bound on the generalization error.
>
> **3. Empirical results are inadequate. Test on large-scale datasets like ImageNet-1K is common in SAM-related works. That would be better if there were more experiments on other tasks like noisy label task.** Our main goal is to expand the existing theories on understanding sharpness-aware minimization (SAM) and to bridge the substantial gap between current theory and practice, rather than to propose a highly practical, and very scalable optimizer to replace SAM. Therefore, we did not conduct experiments on large datasets, such as ImageNet. Nevertheless, in the rebuttal PDF, we show the results of fine-tuning on CIFAR-10/CIFAR-100 using ViT-B-16 pretrained on ImageNet, showing that Eigen-SAM consistently outperforms original SAM. Additionally, we did not conduct experiments related to label noise because we have not developed a theory concerning label noise.
>
> **4. Eigen-SAM requires extra computational cost for estimating the top eigenvector. While the authors claim that it only needs 5%-10% overhead, is the overhead related to the size of model architectures? Whether the estimation need more frequent if tested on large-scale datasets?** Our additional overhead comes from running the Hessian-vector product p times every k steps to estimate the top eigenvector. The Hessian-vector product roughly takes 1-2 times the time required to compute the gradient, so the overhead of our algorithm is approximately $(2+p/k)$ to $(2+2p/k)$ times that of SGD, compared to 2 times for standard SAM. For larger models, the computation time for the Hessian-vector product hardly increases. We refer to [1] for a more detailed analysis of the computation cost of Hessian-vector products. In Table 2 and Table 3 of our rebuttal PDF, we show how larger values of k affect generalization performance and observe that even setting k to 1000 (resulting in less than 1% additional overhead) can retain most of the performance gains. Regarding the question of whether larger models require more frequent estimation of the top eigenvector, we show in Figure 2 of our rebuttal PDF the rate of decay in the quality of the estimated top eigenvector. For larger models, this decay rate only slightly increases and generally decays relatively slowly.
>
> **5. typos** We are very grateful for your careful help in finding them. We deeply apologize for the errors caused by the tight submission deadline, and we will correct all typos and misuses of notation in future versions.
>
> **Questions: Does third-order SDE shown in Figure 2 & 3 estimate the top eigenvector on each mini-batch or once every $k$
>  mini-batch?** In the numerical simulation experiments of the SDE (Figure 2,3), we estimate the eigenvector in each mini-batch.
>
> References
>
> [1] Dagréou, et al., "How to compute Hessian-vector products?", ICLR Blogposts, 2024.

---

> > ### Comment · Reviewer_WYKu · 2024-08-09
> > **Thanks for the rebuttal**
> >
> > Thanks for the detailed rebuttal and explanation. I have no further questions.

---

### Official Review · Reviewer_4ndr · 2024-07-13

**Soundness:** 3
**Presentation:** 3
**Contribution:** 3
**Rating:** 6
**Confidence:** 3

**Summary:**

The authors consider sharpness aware minimization problem and derive third-order SDE governing SGD dynamics.

As a result, they obtained a bound on the generalization error of SAM, showing that SAM trajectories favor flat minimal with the smaller largest eigenvalue of the Hessian, implying that flatter minima can improve generalization.

**Strengths:**

Third-order approximation by SDE of underlying SGD dynamics and explicit bounds. It is well-written and rigorous.

**Weaknesses:**

The full list of assumptions on the loss is not listed. Following the proofs, I can see that it should satisfy uniform boundness and the existence of a non-zero spectral gap w.r.t. the largest eigenvalue of the Hessian. I would appreciate it if the authors specified a list of assumptions and examples of losses that satisfy them.

Bound (4) does not necessarily imply improved generalization. Reading the text 'crucial' etc., creates an impression as if it is. I would appreciate it if it would be mentioned that the paper proposes a possible explanatory mechanism.

**Questions:**

How to practically verify assumptions on the loss? Do the assumptions of the main theorems hold in conducted empirical study?

**Limitations:**

-

---

> ### Author Rebuttal · Authors · 2024-08-07
>
> I appreciate the time and effort you've taken to review our manuscript. Below are our responses to your concerns:
>
> * **The full list of assumptions on the loss is not listed.** The assumptions for Theorem 4.1 can be found in Section A.2 of the appendix, including the continuity of the third derivatives, assumptions on the parameter domain, and the boundedness of the loss function $f$ on the parameter domain. The assumptions for Theorem 4.2 and Corollary 4.2.1 can be found in Assumption 1 in Appendix A.1, which include the continuous differentiability of $f$, at most polynomial growth rate, and Lipschitz conditions. We will make these lists of assumptions more clearly visible in future versions.
>
> * **Bound (4) does not necessarily imply improved generalization.** We need to clarify that bound (4) is an upper bound on the generalization error, which is used to link the largest eigenvalue of the Hessian matrix with the generalization error. Even though there is a correlation, we acknowledge that this does not mean a smaller eigenvalue necessarily implies a smaller generalization error. In practice, we rely on our proposed algorithm, Eigen-SAM, to regularize the largest eigenvalue rather than directly optimizing the bound on the generalization error.
>
> * **How to practically verify assumptions on the loss? Do the assumptions of the main theorems hold in conducted empirical study?** Our assumptions are quite mild and do not introduce additional assumptions compared to previous theoretical work. We refer you to some prior empirical studies (e.g. [1], [2]) that demonstrate how these assumptions can be validated in practice, but this is beyond the scope of this paper.
>
> References
>
> [1] Fazlyab, M., Robey, A., Hassani, H., Morari, M., & Pappas, G. (2019). Efficient and accurate estimation of lipschitz constants for deep neural networks. Advances in neural information processing systems, 32.
>
> [2] Khromov, G., & Singh, S. P. (2024). Some Fundamental Aspects about Lipschitz Continuity of Neural Networks. In The Twelfth International Conference on Learning Representations.

---

### Official Review · Reviewer_eUoc · 2024-07-23

**Soundness:** 3
**Presentation:** 2
**Contribution:** 3
**Rating:** 6
**Confidence:** 5

**Summary:**

### Summary:



The authors first provide a theoretical connection between max e.v. of Hessian and generalization via PAC-Bayesian bounds. Then, they propose a 3rd-order SDE for SAM, which has a lower approximation error compared to existing 2nd-order SDEs in the literature. They argue that the perturbation-eigenvector alignment is important, so they propose the Eigen-SAM algorithm which estimates max e.v. and improves alignment using power method. The paper concludes with experiments showing improvement over SAM and SGD.

### Main Results:
- Theorem 4.1: relating max e.v. of Hessian to generalization via PAC-Bayesian bounds.
- Theorem 4.2: a 3rd order SDE for SAM, that is used later to study top e.v. alignment.

**Strengths:**

### Pros:

- the paper is super easy to read
- notations are well defined, the problem is clearly formulated, and the results are explicitly stated


- provides a deeper understanding of SAM convergence and implicit bias

**Weaknesses:**

### Cons:

- the need for 3rd order SDEs for SAM are not well justified
- the current version can be improved in terms of clarity/typos

**Questions:**

### Questions/Comments:


This is an interesting paper with improvements over the previous works on understanding SAM dynamics and implicit bias. I'm still not well convinced about the need for 3rd order data for SAM analysis. The reuslts are interesting but the paper is lacking a clear introduction/motivation for this study. Also, the provided Eigen-SAM algorithm looks extensive since it relies on Hessian-vector products. The limitations should be discussed clearer in the early pages of the paper.


My comments are as follows:

- I recommend rewritting Theorem 4.1 with exactly parsing the role of each term and also redefining some key quantities there. For example, what is $N$? I believe it is the number of samples (based on previous PAC-Bayesian bounds) but nothing explained there.

- It's worth mentioning that Theorem 4.1 does not work for over parameterized models (as expected). Becasue to make the upper bound smaller one needs to have a very large number of samples.




#### Other comments:




- line 79 -- I suggest using $\|$ instead of $||$ to denote norms.

- line 81 -- is $\nabla^3 f(x)(u,v)$ a vector? Is it a multilinear function? How does it relate to the symmetric tensor $\nabla^3 f(x)$? This part of notations is a bit unclear.


- line 84 -- It might be useful to emphasize that you use the 2-norm of vectors to define SAM.

- line 85 -- typo "aslo"

- line 92 -- the notation here is a bit weird, though I can totally understand what you mean that. I don't have any suggestion right now but it can be confusing for unfamiliar readers.

- line 156 -- "SDE 5" and "SDE 3" are using confusing notation.

- line 160 -- typo $x_\in$

- line 163 -- Is the expectation over the randomness of $X_t$? Then, it just adds some constants to $f(X_t)$ (i.e., independent of $X_t$) and then will vanish in Equation (5)? I think $X_t$ must be replaced with $x$ and then in Equation (5) $x$ is replaced with $X_t$?

- line 163 -- why $\Sigma^{2,2}$ does not depend on $x$? Is this a typo?

- line 167 -- I suggest replacing $g \in G$ with just a few words explaining it. This makes that part clearler.

- line 195 -- it looks like to be typo, what does the middle term mean in the formula?

- Experiments in Figure 1 look noisy -- how many times did you iterate it?

- line 290 -- typo

- How did Algorithm 1 estimate the top eigenvector? Can you explain this with a simple example? I mean, the power method in words/quick examples.

- Missing references: I suggest discussing the following two related papers in the future version of the paper. One of them is related to random perturbations, decomposition, and trace regularization, and the other one is related to SAM algorithms with general sharpness measures:
        - A Universal Class of Sharpness-Aware Minimization Algorithms, ICML 2024
        - How to escape sharp minima with random perturbations, ICML 2024

---

> ### Author Rebuttal · Authors · 2024-08-07
>
> Thank you for your thorough and insightful feedback, which has been invaluable to the improvement of our work. Below we address your comments:
>
> * **line 81 -- is $\nabla^3f(x)(u,v)$ a vector? Is it a multilinear function? How does it relate to the symmetric tensor $\nabla^3f(x)$?** Yes $\nabla^3f(x)(u,v)$ is a vector with the same size with $x$, it can be written as $(\nabla^3f(x)(u,v))_k=u^T(\nabla^3f(x))_kv$, it is bilinear with respect to the variables $u,v$.
>
> * **Suggestions regarding Theorem 4.1** We appreciate your suggestions regarding the presentation of Theorem 4.1. We will rephrase it in future versions to make it more rigorous and readable. You mentioned that Theorem 4.1 does not work for over parameterized models. It is correct. Because we need to bound the KL divergence between the prior and the posterior, dependence on the number of parameters in the bound is unavoidable if we do not impose additional constraints on the prior and posterior. The reason we did not choose to add more constraints is that we wanted to remain consistent with other PAC-Bayes type bounds related to SAM (e.g. [1], [2], [3]).
>
> * **line 163 -- Is the expectation over the randomness of $X_t$? Then, it just adds some constants to $f(X_t)$ and then will vanish in Equation(5)? I think $X_t$ must be replaced with $x$ and then in Equation (5) is replaced with $X_t$?** No, we mentioned in Section 3.1 when introducing the notation that unless otherwise specified, our expectations are with respect to minibatch $\gamma$. Therefore, $f(X_t)$ is still a function of $X_t$. $X_t$ and $x_k$ are different stochastic processes, the former is the solution of the SDE, while the latter is obtained by running the discrete algorithm.
>
> * **line 163 -- why $\Sigma^{2,2}$ does not depend on $x$? Is this a typo?** Yes, it is a typo. In fact, it should depend on $X_t$.
>
> * **line 195 -- it looks like to be typo, what does the middle term mean in the formula?** It is not a typo. $s^*$ and $\lambda_1$ are both scalars. We will rewrite this expression to make it more readable.
>
> * **Experiments in Figure 1 look noisy -- how many times did you iterate it?** We did not run repeated experiments because we only aimed to obtain a qualitative result, and the overall trend is clear.
>
> * **How did Algorithm 1 estimate the top eigenvector? Can you explain this with a simple example? I mean, the power method in words/quick examples.** The power method is an iterative technique used to find the dominant eigenvalue and corresponding eigenvector of a square matrix. It starts with a random unit vector $v$, at every iteration, the vector $v$ is multiplied by the matrix and normalized. The convergence is geometric, with ratio $\lambda_1/\lambda_2$. We refer to Figure 1 in our rebuttal PDF, where we demonstrate the convergence speed of this algorithm in practice.
>
> * **Missing references** We will cite the relevant and interesting concurrent work you mentioned in future versions.
>
> * **All typos and misuses of notation** We are very grateful for your careful help in finding them. We deeply apologize for the errors caused by the tight submission deadline, and we will correct all typos and misuses of notation in future versions.
>
> References
>
> [1] P. Foret, A. Kleiner, H. Mobahi, and B. Neyshabur, "Sharpness-Aware Minimization for Efficiently Improving Generalization," in *Proc. Int. Conf. Learning Representations (ICLR)*, 2021, doi: 10.48550/arXiv.2010.01412.
>
> [2] Zhuang, J., Gong, B., Yuan, L., Cui, Y., Adam, H., Dvornek, N., ... & Liu, T. (2022). Surrogate gap minimization improves sharpness-aware training. arXiv preprint arXiv:2203.08065.
>
> [3] Li, B., & Giannakis, G. (2024). Enhancing sharpness-aware optimization through variance suppression. Advances in Neural Information Processing Systems, 36.

---

> > ### Comment · Reviewer_eUoc · 2024-08-12
> >
> > Thank you for your response and for addressing my comments. I strongly recommend the authors to add the explanation on the power method to their paper. Moreover, please replace Figure 1 with new experiments showing less noisy observations (e.g., by repeating the experiments and taking averages).
> >
> > Given that most of my comments have been addressed, I have decided to increase the score of this paper.

---

> ### Author Response · Authors · 2024-08-12
>
> Thank you for your positive feedback and for increasing the score. We will incorporate the explanation of the power method and update Figure 1 with averaged results from repeated experiments to reduce noise.

---

### Author Rebuttal · Authors · 2024-08-07

We would like to thank all reviewers for their careful and thoughtful feedback. Some concerns from the reviewers are centered on our experimental section, therefore, we would like to provide a general response here. Firstly, we would like to clarify that our paper is more theoretical in nature. Our main goal is to expand the existing theories on understanding sharpness-aware minimization (SAM) and to bridge the substantial gap between current theory and practice, rather than to propose a highly practical, and very scalable optimizer to replace SAM. Nevertheless, we still hope to provide some supporting experiments in our rebuttal PDF to further support our algorithm, including: (1) Fine-tuning on CIFAR-10/CIFAR-100 using ViT-B-16 pretrained on ImageNet, where we have presented the test accuracies in Table 1, showing that Eigen-SAM consistently outperforms original SAM; (2) Conducting a sensitivity analysis on the hyperparameter k (interval steps for estimating eigenvectors), with the experimental results reported in Tables 2 and 3. We found that setting k=1000, while the additional computational cost is less than 1%, retains most of the performance gain; (3) Showing the convergence rate of Algorithm 1 used for estimating the top eigenvalue in Figure 1; (4) Reporting the rate of decay in the quality of the estimated top eigenvectors as training progresses in Figure 2. We hope these experiments will adequately support the effectiveness of our algorithm and address some of the reviewers' concerns. We will respond individually to each reviewer's specific issues in the rebuttals below. Thank you for your patience.

---

### Decision · Program_Chairs · 2024-09-25

**Decision:**

Accept (poster)

**Comment:**

The authors consider sharpness aware minimization problem and derive third-order SDE governing SGD dynamics. All authors agreed on the theoretical contribution in the paper, while the experiments could be improved. Reviewers and authors conducted constructive discussions and I believe that this paper is a useful contribution to deep learning optimization.